# CaPulse: Detecting Anomalies by Tuning in to the Causal Rhythms of Time Series

## Abstract

Time series anomaly detection has garnered considerable attention across diverse domains. While existing methods often fail to capture the underlying mechanisms behind anomaly generation in time series data. In addition, time series anomaly detection often faces several data-related inherent challenges, i.e., label scarcity, data imbalance, and complex multi-periodicity. In this paper, we leverage causal tools and introduce a new causality-based framework termed **CaPulse**, which "tunes in" to the underlying "causal pulse" of time series data to effectively detect anomalies. Concretely, we begin by building a structural causal model to decipher the generation processes behind anomalies. To tackle the challenges posed by the data, we propose Periodical Normalizing Flows with a novel mask mechanism and carefully designed periodical learners, creating a periodicity-aware, density-based anomaly detection approach. Extensive experiments on seven real-world datasets demonstrate that CaPulse consistently outperforms existing methods, achieving AUROC improvements of 3% to 17%, with enhanced interpretability. Our source code is available at this anonymized repository: `https://anonymous.4open.science/r/iclr25-s622/aiops/CaTAD/README.md`.

## 1 Introduction

Time Series Anomaly Detection (TSAD) has gained significant attention in recent years (Darban et al., 2024) due to its applications across diverse domains such as network security (Ahmed et al., 2016), finance (Takahashi et al., 2019), urban management (Bawaneh & Simon, 2019), and cloud computing services (Ren et al., 2019; Chen et al., 2024a). Traditional TSAD methods, including one-class support vector machines (Schölkopf et al., 2001) and kernel density estimation (Kim & Scott, 2012), rely heavily on handcrafted features and struggle to handle high-dimensional time series data effectively. In contrast, Deep Learning (DL)-based approaches have recently emerged, significantly improving detection performance thanks to their powerful representation learning capabilities (Ruff et al., 2018; Sabokrou et al., 2018; Goyal et al., 2020).

Despite their promise, DL-based methods for TSAD face several key limitations. ***Mechanistically***, they often overlook the underlying patterns and processes behind anomalies generation in time series data, leading to models that lack interpretability and exhibit limited generalization capabilities. Causal inference (Pearl et al., 2000) provides a powerful platform for investigating the underlying causal systems, with successful integration in DL methods demonstrated across various tasks (Lv et al., 2022; Zhao & Zhang, 2024). Specifically, by incorporating a *causal perspective*, models can uncover the true factors driving anomalies, rather than relying solely on statistical dependencies or superficial correlations. This shift toward causal-based methods not only improves generalization – making models more robust in Out-of-Distribution (OoD) scenarios (Yang et al., 2022) – but also significantly enhances interpretability, providing deeper insights into the root causes of anomalies. This is particularly essential for downstream tasks such as root cause analysis, where pinpointing the specific factor responsible for an anomaly, like identifying a server overheating or a hardware malfunction causing system downtime in cloud services, becomes critical for timely and effective intervention (Li et al., 2022). Yet, there is still substantial potential for further exploration of causal methods in TSAD.

In addition to the mechanical aspect, ***intrinsically***, anomaly detection in time series is challenged by three critical characteristics in terms of data themselves: *label scarcity*, *data imbalance*, and *multiple periodicities*. In practice, e.g., in industrial systems or cloud computing platforms (Zhang et al., 2024), acquiring labeled anomalies is often impractical due to the significant manual effort and cost required (Chen et al., 2024b). Even when labels are available, datasets typically consist of

both normal and anomalous instances, which can result in overfitting to noisy labels (Wang et al., 2019; Huyan et al., 2021) and degrading model performance (Zhou et al., 2023b) (Figure 1a). Additionally, many time series exhibit multiple periodicities, with short-term cycles, e.g., hourly or daily fluctuations, overlapping with long-term patterns that develop over weeks (Wen et al., 2021; Wu et al., 2023). We refer to them as *local* and *global* periodicities, respectively, shown in Figure 1c. This adds complexity to TSAD efforts: in cloud computing services, user misoperations often cause transient anomalies linked to short-term fluctuations, whereas long-term patterns typically signal machine failures or system degradation. However, existing TSAD methods fail to effectively address all three challenges simultaneously, underscoring the need for more advanced solutions.

In this paper, we aim to decipher the underlying generation process of anomalies in time series and provide solutions while addressing the challenges inherent to TSAD, i.e., label scarcity, data imbalance, and multiple periodicities. Concretely, we begin by adopting a causal standpoint, introducing a Structural Causal Model (SCM) (Pearl et al., 2000) to gain a deeper understanding of the causal mechanisms governing anomalies in time series. Building on this, we develop a novel deep learning framework that integrates causality-based solutions for accurate and interpretable TSAD. Meanwhile, motivated by the demonstrated success of density estimation in unsupervised anomaly detection (Rudolph et al., 2021; Gudovskiy et al., 2022; Dai & Chen, 2022) (Figure 1b), we develop a periodicity-aware, density-based approach that effectively addresses three inherent challenges in TSAD. Our contributions are summarized as follows:

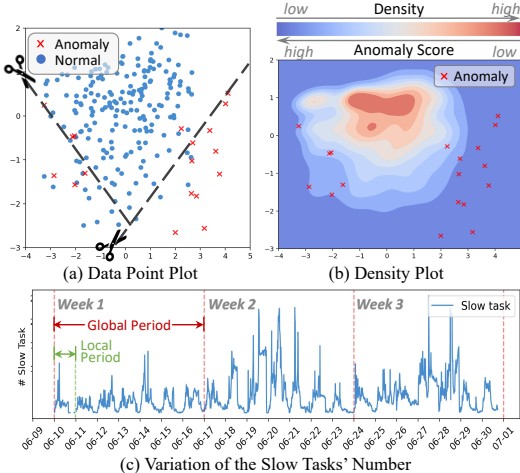

Figure 1: (a) Data point plot and (b) density plot (c) multiple periodicities in the Cloud-S dataset.

- **A causal view for TSAD.** To uncover the underlying generation mechanisms driving anomalies in time series data, we present a causal perspective and propose an Structural Causal Model for the TSAD problem. Building on this, we leverage causal tools to introduce a new framework, **CaPulse**, which *listens to the "pulse" of time series data – its underlying "causal" rhythms –and identifies when something is out of sync.* Like a capsule, CaPulse serves as an anomaly detector by pinpointing the true underlying issues in the time series.

- **A novel periodicity-aware density-based approach.** To tackle the intrinsic challenges of data, we propose Periodical Normalizing Flows, which are built on conditioned normalizing flows to enable *unsupervised density-based* anomaly detection. For capturing *multi-period* dynamics, CaPulse integrates different periods' local information by learning causal pyramid representations as conditioning inputs, and global period information is incorporated via a novel mask mechanism.

- **Comphrehensive empirical evidence.** We validate the effectiveness of CaPulse through extensive experiments on seven real-world datasets spanning five different domains. The results show that the proposed model consistently outperforms existing baselines on most datasets, achieving AUROC improvements ranging from 3% to 17%, while also providing clearer interpretability.

## 2 PRELIMINARIES

### 2.1 PROBLEM STATEMENT

In this paper, we focus on *unsupervised* anomaly detection in multivariate time series data. Let $\boldsymbol{X}^{1:T} = \{\boldsymbol{x}_1, \cdots, \boldsymbol{x}_T\} \in \mathbb{R}^{T \times D}$ represent multivariate time series, each $\boldsymbol{x}_t \in \mathbb{R}^D$ denotes the data at time point $t$, where $T$ is the length of the time series, and $D$ is the dimensionality. For a given $\boldsymbol{X}^{1:T}$, our target is to yield anomaly scores for all time points, denoted as $\boldsymbol{\tau}^{1:T} = \{\tau_1, \cdots, \tau_T\} \in \mathbb{R}^T$, where each $\tau_t \in \mathbb{R}$ indicates the likelihood of an anomaly at time $t$. For evaluation, we consider a corresponding set of labels $\boldsymbol{y}^{1:T} = \{y_1, \cdots, y_T\} \in \mathbb{R}^T$, where $y_t \in \{0, 1\}$ indicates whether a time point is normal ($y_t = 0$) or anomalous ($y_t = 1$). For conciseness, we refer to $\boldsymbol{X}^{1:T}$ as $\boldsymbol{X}$, $\boldsymbol{y}^{1:T}$ as $\boldsymbol{y}$, and $\boldsymbol{\tau}^{1:T}$ as $\boldsymbol{\tau}$ in the rest of the paper.

## 2.2 Related Works

**Time Series Anomaly Detection** (TSAD) has advanced from traditional statistical methods (McLachlan & Basford, 1988; Schölkopf et al., 1999; Breunig et al., 2000; Tax & Duin, 2004) to complex Deep Learning (DL) methods (Schmidl et al., 2022; Darban et al., 2024). While DL methods such as forecasting- (Hundman et al., 2018; Shen et al., 2020) and reconstruction-based models (Su et al., 2019; Audibert et al., 2020; Xu et al., 2022) offer improved detection, they can struggle with rapidly changing data and noisy labels (Golestani & Gras, 2014; Zhou et al., 2023b; Chen et al., 2024b). Density-based methods (Dai & Chen, 2022; Zhou et al., 2023b) provide robust performance across scenarios. Recently, large-scale pre-trained models such as AnomalyLLM (Liu et al., 2024) and AnomalyBERT (Jeong et al., 2023) have emerged. Yet, most methods focus on statistical dependencies, often overlooking the underlying generation process behind anomalies.

**Causal Inference** (CI) (Pearl et al., 2000; Glymour et al., 2016) seeks to investigate causal relationships between variables, ensuring robust learning and inference. Integrating DL techniques with CI has shown great promise in recent years, especially in computer vision (Zhang et al., 2020; Lv et al., 2022), natural language processing (Roberts et al., 2020; Tian et al., 2022), and spatio-temporal data mining (Xia et al., 2023; Wang et al., 2024). In the realm of sequential data, CI is often leveraged to address temporal OoD issues by learning disentangled seasonal-trend (Woo et al., 2022) or environment-specific representations (Yang et al., 2022) to enhance forecasting accuracy. Though promising, the intrinsic causal mechanisms in TSAD differ from the prediction problem, and the application of CI in this domain remains in its early stages.

**Normalizing Flows** (NFs) (Tabak & Turner, 2013; Papamakarios et al., 2021) are a powerful technique for density estimation, widely applied in tasks such as image generation (Papamakarios et al., 2017). Advanced variants have been developed to enhance models' applicability, such as Real-NVP (Dinh et al., 2017). Recently, NFs have been explored for anomaly detection across many domains, relying on the assumption that anomalies reside in low-density regions (Rudolph et al., 2021; Gudovskiy et al., 2022). In the time series realm, following an initial application of NFs for time series forecasting (Rasul et al., 2021), NFs-based TSAD has been explored by GANF (Dai & Chen, 2022). Following this, MTGFlow (Zhou et al., 2023b) further improves model capacity through an entity-aware NFs design. However, these methods fail to account for the multiple periodicities inherent in time series and overlook the generative processes that give rise to anomalies.

## 3 A Causal View on TSAD

### 3.1 Causal Perspective: Generation of Anomalies

Existing TSAD methods typically infer anomalies $\boldsymbol{y}$ based solely on the input time series $\boldsymbol{X}$, i.e., by modeling $P_\theta(\boldsymbol{y}|\boldsymbol{X})$, as shown in Figure 2a, where $P_\theta(\cdot)$ denotes the distribution induced by a model $f_\theta$. However, real-world scenarios are often more complex than modeling these statistical dependencies between the input $\boldsymbol{X}$ and the label $\boldsymbol{y}$, since there exist various underlying factors directly or indirectly influencing the anomaly generation. To address this, we adopt a causal perspective and introduce a Structural Causal Model (SCM) (Pearl et al., 2000) to describe the anomaly generative process in time series, as illustrated in Figure 2b, aiming to uncover the intrinsic causal relationships between different variables in the context of TSAD. Rather than solely modeling the statistical dependency $P_\theta(\boldsymbol{y}|\boldsymbol{X})$, we propose focusing on $P_\theta(\boldsymbol{y}|do(\boldsymbol{U}),\boldsymbol{C})$.

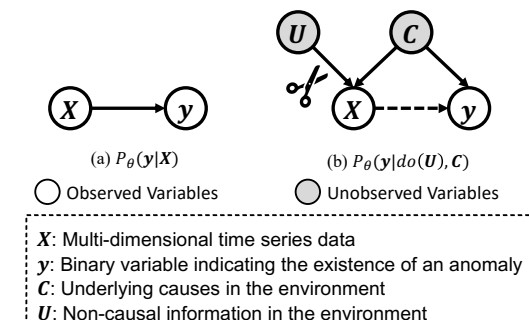

(a) $P_\theta(\boldsymbol{y}|\boldsymbol{X})$      (b) $P_\theta(\boldsymbol{y}|do(\boldsymbol{U}),\boldsymbol{C})$

○ Observed Variables    ● Unobserved Variables

$\boldsymbol{X}$: Multi-dimensional time series data
$\boldsymbol{y}$: Binary variable indicating the existence of an anomaly
$\boldsymbol{C}$: Underlying causes in the environment
$\boldsymbol{U}$: Non-causal information in the environment

Figure 2: SCMs of (a) Existing TSAD methods; (b) Time series anomaly generation under real-world scenarios. Solid arrow: causal relationships. Dash arrow: statistical dependencies. Scissors: causal intervention $do(\cdot)$.

To facilitate understanding, consider an example of cloud computing services. Here, the sequential data $\boldsymbol{X}$ includes the number of slow tasks running on a specific instance within the cloud platform, CPU memory usage, allocated compute resources, and other related metrics that evolve over time. Our goal is to identify issues or anomalies $\boldsymbol{y}$ within this instance caused by true underlying causal

factors $C$ from the environment. Here $C = \{c_1, c_2, \cdots, c_N\} \in \mathbb{R}^{N \times D_c}$ refers to all latent causal factors, such as "*hardware failures*" and "*network latency*". $N$ and $D_c$ refer to the number and the dimensions of causal factors, respectively. Yet, there are some non-causal factors $U$ also in the environment, such as "*user mis-operations*" or "*data collection jitter*", which may affect the readings of $X$ but do not impact the instance itself, thus unrelated to our detection goal $y$. Thus, an ideal detector is expected to root out the influence of $U$ and focus solely on the causal part $C$. More discussion and another example can be found in Appendix G.1.

### 3.2 CAUSAL BACKING: INDEPENDENCE REQUIREMENT

Based on the SCM in Figure 2b, our aim is to detect anomalies by identifying their true underlying causes while eliminating the influence of non-causal factors, i.e., modeling $P_\theta(y|do(U), C)$. The $do(\cdot)$ operator, as defined in do-calculus, signifies an intervention on the variable (Glymour et al., 2016). Directly modeling this operator is challenging because it necessitates learning various latent causes $C$ from the raw input $X$ (Arjovsky et al., 2019). Inspired by a previous work (Lv et al., 2022), we alternatively leverage a couple of widely-used principles from the causal theory to force the representation of causal factors $C$ we learned to satisfy following key properties.

**Common Cause Principle** (Reichenbach, 1991) posits that for two statistically dependent variables $X$ and $Y$, there exists a variable $C$ that causally influences both, thereby explaining their dependence by rendering them conditionally independent when conditioned on $C$. Accordingly, the SCM depicted in Figure 2b can be formalized as $X := f(C, U)$ and $y := h(C) = h(g(X))$, where $C \perp\!\!\!\perp U$. Here, $f$, $h$, and $g$ denote unknown structural functions that describe how the observed variables $X$ and $y$ are generated from the underlying causes $C$ and the non-causal variable $U$. This leads to our first property for $C$: it should be independent of $U$. In this way, for any distribution $P(X, y)$, given the causal factor $C$, there exists a conditional distribution $P(y|C)$ that represents the invariant mechanism triggering the anomaly within time series.

**Independent Causal Mechanisms** (Schölkopf et al., 2012; Peters et al., 2017) suggest that the conditional distribution of each variable, given its causes, does not influence other causal mechanisms. In other words, none of the factorization of $C$ entails information of others (Lv et al., 2022). Thus it enforces the mutual independence of the causal factors $C = \{c_1, c_2, \ldots, c_N\}$, where $N$ is the number of latent causal factors.

Therefore, instead of directly learning the causal factors $C$, we enforce them to satisfy the following requirements: **R1)** $C$ should be independent of $U$, i.e., $C \perp\!\!\!\perp U$, and **R2)** the components of $C$ should be mutually independent, i.e., $c_1 \perp\!\!\!\perp c_2 \perp\!\!\!\perp \ldots \perp\!\!\!\perp c_N$.

## 4 MODEL INSTANTIATIONS

To implement these causal independent requirements into practice, we develop a causality-based TSAD framework termed **CaPulse** to tune in to the underlying causal rhythms of time series data, as illustrated in Figure 3. Meanwhile, in light of the common challenges in TSAD, we carefully designed the modules in CaPulse to enable it not only to effectively implement these causal solutions but also to address key data-related challenges. In this section, we begin with an overview of the proposed framework, followed by a detailed explanation of each component.

**Overview.** The input time series $X$ is first augmented to generate $X'$ via adding noise on their high-frequency components. Both $X$ and $X'$ are subsequently passed through the Periodicity-aware Cause Miner (PaFM) module to obtain $C_p$ and $C'_p$, i.e., the pyramid representations of latent causal factors at different frequencies. PaFM also outputs the amplitude weights for each frequency, denoted as $w_p$ and $w'_p$. Next, the Multi-period Cause Fusion (MpCF) module fuses information across different periods based on the amplitude weights and a plugged attention mechanism to generate the omni representations $C_o$ and $C'_o$. A similarity loss $\mathcal{L}_{sim}$ ensures the consistency between these two representations. Then the final representation $C_{ind}$ is obtained by averaging them. To ensure the independence of the learned causal factors, we impose an orthogonal loss $\mathcal{L}_{ind}$. After that, Periodical Normalizing Flows (PeNF) takes $X$, the global period $p_g$ (obtained by Fast Fourier Transform), and $C_{ind}$ as inputs to estimate the density of $X$ by learning a sequence of invertible transformations, mapping $X$ into a simpler distribution $P(Z)$, optimized through the loss $\mathcal{L}_{nf}$.

Figure 3: The pipeline of CaPulse. Different color shaded areas denote solutions for causal treatments , multiple periodicities , and label scarcity & data imbalance , respectively. Ind.: Independent. ICM: Independent Causal Mechanisms.

## 4.1 CAUSAL TREATMENTS

**Causal Intervention.** Since $C$ should be separated from $U$ (**R1**), performing an intervention upon $U$ does not make changes to $C$. We thus leverage causal intervention $do(\cdot)$ (Pearl et al., 2000), to mitigate the negative influence of non-causal factors $U$ and to extract causal representations $C$ that are unaffected by $U$ (Lv et al., 2022; Zhou et al., 2023a). In real-world scenarios, non-causal elements such as user misreports often occur randomly, akin to noise typically found in the high-frequency components of time series data. Following insights from prior researches (Gao et al., 2021; Xia et al., 2024), we conduct causal intervention by adding noise to the less significant part — the high-frequency components — of the input data to simulate real-world disturbances. This process is formalized as follows:

$$X' = \text{iFFT}(\text{concat}[\text{FFT}(X)_{0:k_h}, \text{FFT}(X)_{k_h:T} + \eta]), \tag{1}$$

where $\text{FFT}(\cdot)$ and $\text{iFFT}(\cdot)$ denote the Fast Fourier Transform, a commonly-used method for separating high- and low-frequency components, and its inverse. $\text{FFT}(\cdot)_{i:j}$ denotes the $i$-th to $j$-th components, $k_h$ refers to the high-frequency threshold, and $\eta \sim \mathcal{N}(0, \sigma^2)$ is the added noise. Then we obtain the causal representations $C_o$ and $C'_o \in \mathbb{R}^{N \times D_c}$ of $X$ and $X'$ via PaCM and MpCM modules. To ensure the learned information only contains the invariant causal part, we enforce consistency in them and minimize their difference via a similarity loss $\mathcal{L}_{\text{sim}} = \frac{\langle C_o, C'_o \rangle}{\|C_o\| \|C'_o\|}$. Details on PaCM and MpCF will be introduced in Section 4.2.

**Joint Independence**. After obtaining $C_o$ and $C'_o$, the final causal representation $C_{\text{ind}}$ is computed by averaging the two variables. To enforce the joint independence requirement (**R2**), we apply an orthogonal loss that penalizes deviations from independence. This is achieved by measuring the squared Frobenius norm of the difference between $C_{\text{ind}}^\top C_{\text{ind}}$ and the identity matrix $I$: $\mathcal{L}_{\text{ind}} = \left\| C_{\text{ind}}^\top C_{\text{ind}} - I \right\|_F^2$. This loss encourages the dimensions of $C_{\text{ind}}$ to be orthogonal, ensuring that the extracted causal factors are distinct and independent.

## 4.2 MULTI-PERIODICITY AWARENESS

Then we detail the capture of the local and global periodic information (the orange part in Figure 3).

**Local Periodical Pyramid & Fusion**. We introduce the PaCM module to extract causal factors for $k$ periodicities, denoted as $\mathbf{C}_p = \{C_{p1}, C_{p2}, \ldots, C_{pk}\} \in \mathbb{R}^{N \times D_h \times k}$, along with their corresponding amplitudes $w_p = \{w_{p1}, w_{p2}, \ldots, w_{pk}\} \mathbb{R}^k$. Inspired by TimesNet (Wu et al., 2023), this module transforms the input into the frequency domain, selects the top $k$ frequency periods, and reshapes them based on their periodicity. The MpCF module then applies self-attention to compute attention scores $a_p = \{a_{p1}, a_{p2}, \ldots, a_{pk}\} \in \mathbb{R}^k$ for each period. After that, it aggregates variables of different periods using both $w_p$ and $a_p$ to generate the final omni representation $C_o \in \mathbb{R}^{N \times D_h}$. The attention mechanism in MpCF dynamically adjusts the importance of each periodic component based on their interactions and dependencies within the time series, rather than relying solely on amplitude when fusing the information across periods. This also enhances the interpretability of the model, discussed in Section 5.3. Due to space constraints, details of PaCM and MpCF are provided in Appendix B.1.

(a) Period Checkerboard Mask (PC-Mask)  (b) Periodical Normalizing Flows (PeNFs)

Figure 4: (a) Masking schemes PC-Mask. (b) Architecture of PeNF, where the black and red arrows represent the data flow for the input and the conditional variable, respectively.

**Global Periodical Checkerboard Mask.** To enhance the model's global period awareness, we introduce the PC-Mask scheme tailored to the proposed PeNF, illustrated in Figure 4. First, for the total length $T_l$ time series with $D$ dimensions $\boldsymbol{X}^{1:T_l}$, we discover the global period $p_g$ as follows:

$$\boldsymbol{a} = \mathrm{Avg}\left(\mathrm{Amp}\left(\mathrm{FFT}(\boldsymbol{X}^{1:T_l})\right)\right), \ f_g = \arg\max\left(\boldsymbol{a}\right), \ p_g = \left\lceil \frac{T_l}{f_g} \right\rceil. \tag{2}$$

Here, $\mathrm{Amp}(\cdot)$ and $\mathrm{Avg}(\cdot)$ denotes the average calculation of amplitude values. $\boldsymbol{a} \in \mathbb{R}^{T_l}$ represents the averaged amplitude of each frequency. The $j$-th value $\boldsymbol{a}_j$ represents the intensity of the frequency-$j$ periodic basis function, corresponding to the period length $\lceil \frac{T_l}{j} \rceil$. We select the largest amplitude values to obtain the most significant frequencies $f_g$, and then we regard its corresponding period length $p_g$ as our global period. Next, we use $p_g$ to create PC-Mask $\boldsymbol{M} \in \mathbb{R}^{T \times D}$ by a repeating pattern of $p_g$ zeros followed by $p_g$ ones, illustrated in Figure 4a. This process is formulated as $m_j^i = \left(\left\lfloor \frac{j}{p_g} \right\rfloor \mod 2\right)$, where $m_j^i$ is the element of the mask $\boldsymbol{M}$ at position $(i, j)$, $\lfloor \cdot \rfloor$ denotes the floor function, which returns the greatest integer less than or equal to the input, and $\mod$ denotes the modulo operation. This mask will be used for a periodicity-aware density estimation for anomaly detection, detailed in the following section.

### 4.3 DENSITY ESTIMATION

To address the issue of limited labels and imbalanced data, we leverage NFs to achieve an unsupervised density-based anomaly detector. Building on the success of conditioned NFs for time series (Rasul et al., 2021), we propose PeNF (Figure 4b), augmenting conditioned NFs with periodically-awareness introduced by the PC-Mask mechanism. Overall, PeNF performs the density estimation of the input $\boldsymbol{X}$ conditioned on the learned causal representation $\boldsymbol{C}_{\mathrm{ind}}$ by learning a sequence of invertible functions $\mathcal{F}$ mapping $\boldsymbol{X}$ into a simple distribution $P(\boldsymbol{Z})$. With the flows parameterized with $\theta$, i.e., $\mathcal{F}_\theta : \mathbb{R}^D \times \mathbb{R}^{D_h} \to \mathbb{R}^D$, where $D_h$ denotes the hidden dimension, the conditioned distribution of $\boldsymbol{X}$ can be expressed as:

$$P_\mathcal{X}(\boldsymbol{X}|\boldsymbol{C}_{\mathrm{ind}}) = P_\mathcal{Z}(\boldsymbol{Z}|\boldsymbol{C}_{\mathrm{ind}})\left|\det \frac{\partial \boldsymbol{Z}}{\partial \boldsymbol{X}}\right| = P_\mathcal{Z}(\mathcal{F}_\theta(\boldsymbol{X}, \boldsymbol{C}_{\mathrm{ind}}))\left|\det \frac{\partial \mathcal{F}_\theta(\boldsymbol{X}, \boldsymbol{C}_{\mathrm{ind}})}{\partial \boldsymbol{X}}\right|, \tag{3}$$

where $|\det(\partial \mathcal{F}_\theta/\partial \boldsymbol{X})|$ is the Jacobian of $\mathcal{F}_\theta$ at $\boldsymbol{X}$ and $P_\mathcal{Z}$ is the distribution of $\boldsymbol{Z} \in \mathbb{R}^{T \times D}$ which is chosen to be the standard normal $\boldsymbol{z} \sim \mathcal{N}(0, \boldsymbol{I}) \in \mathbb{R}$ in this work.

In practice, PeNF takes the PC-Mask $\boldsymbol{M}$ (or the global period $p_g$), the causal representation $\boldsymbol{C}_{\mathrm{ind}}$ and the input data $\boldsymbol{X}$ as its input. Inspired by Dinh et al. (2017) and Rasul et al. (2021), we design a *periodic contextual layer* to enable NFs aware of periodicity and PeNF consists of $L$ periodic contextual layers. In the $l$-th layer, there are two outputs: $\boldsymbol{H}_l$ and $\boldsymbol{J}_l$. The first output will be passed to the next layer for further updates, while the second output will be accumulated across layers and contribute to the final Jacobian variable $\log|\det(\partial \mathcal{F}_\theta/\partial \boldsymbol{X})|$, which be used for optimize and detailed in the next section. To obtain $\boldsymbol{H}_l$, we use the mask $\boldsymbol{M}$ derived based on the global period $p_g$ to select part of the input $\boldsymbol{H}_{l-1}$ to remain unchanged: $\boldsymbol{H}_{l-1}' = \boldsymbol{H}_{l-1} \odot \boldsymbol{M}$, where $\odot$ denotes the Hadamard product and $\boldsymbol{H}_0 = \boldsymbol{X}$. The remaining part of the input, $\hat{\boldsymbol{H}}_{l-1}' = \boldsymbol{H}_{l-1} \odot (\boldsymbol{I} - \boldsymbol{M})$, is transformed via functions of the unaltered variables. Thus, in the $l$-th layer, $\boldsymbol{H}_l$ will be updated:

$$\boldsymbol{H}_l = \boldsymbol{H}_{l-1}' + (\hat{\boldsymbol{H}}_{l-1}' - \mathcal{T}_\theta(\boldsymbol{H}_{l-1}', \boldsymbol{H}_c) \odot \exp\left(-\mathcal{S}_\theta(\boldsymbol{H}_{l-1}', \boldsymbol{H}_c)\right)), \tag{4}$$

where $\mathcal{S}_\theta(\cdot)$ and $\mathcal{T}_\theta(\cdot)$ are scaling and translation functions parameterized by neural networks with $\theta$, and $\boldsymbol{H}_c \in \mathbb{R}^{T \times D_h}$ is the latent variable obtained by a linear transformation from $\boldsymbol{C}_{\mathrm{ind}}$. Then, a number of these periodic contextual layers mapping are composed together: $\boldsymbol{X} \to \boldsymbol{H}_1 \to \boldsymbol{H}_2 \to \cdots \to \boldsymbol{H}_L \to \boldsymbol{Z}$. More details about normalizing flows can be found in Appendix A.

### 4.4 OPTIMIZATION & ANOMALY MEASUREMENT

Considering the causal independent requirements discussed in Section 4.1, we minimize the total loss: $\mathcal{L} = \mathcal{L}_{\text{nf}} + \alpha\mathcal{L}_{\text{sim}} + \beta\mathcal{L}_{\text{ind}}$, where $\alpha$ and $\beta$ regulate the trade-off of the causal intervention and cause independent loss, and $\mathcal{L}_{\text{nf}}$ is used to optimize the density estimation of $\boldsymbol{X}$ conditioned on $\boldsymbol{C}_{\text{ind}}$, denoted as the negative logarithms of the likelihoods in Eq. 3:

$$\mathcal{L}_{\text{nf}} = -\sum_{t=1}^{T}\left[\log P_{\mathcal{Z}}(\mathcal{F}_{\theta}(\boldsymbol{x}_t, \boldsymbol{c}_t)) + \log\left|\det\frac{\partial\mathcal{F}_{\theta}(\boldsymbol{x}_t, \boldsymbol{c}_t)}{\partial\boldsymbol{x}_t}\right|\right]. \tag{5}$$

Density-based approaches act as anomaly detectors based on the widely accepted hypothesis that abnormal instances exhibit lower densities compared to normal ones (Wang et al., 2020; Zhou et al., 2024). Following prior works (Dai & Chen, 2022; Zhou et al., 2023b), we compute the anomaly score $\tau$ as the negative logarithm of the likelihood of the input time series $\boldsymbol{X}$ in Eq. 3:

$$\tau(\boldsymbol{X}) = -\log P_{\mathcal{X}}(\boldsymbol{X}|\boldsymbol{C}_{\text{ind}}) = -(\log P_{\mathcal{Z}}(\mathcal{F}_{\theta}(\boldsymbol{X}, \boldsymbol{C}_{\text{ind}})) + \log\left|\det\frac{\partial\mathcal{F}_{\theta}(\boldsymbol{X}, \boldsymbol{C}_{\text{ind}})}{\partial\boldsymbol{X}}\right|) \tag{6}$$

## 5 EXPERIMENTS

### 5.1 EXPERIMENTAL SETUP

We evaluate CaPulse on seven real-world datasets from different domains, including five commonly used public datasets for TSAD - MSL (Hundman et al., 2018), SMD (Su et al., 2019), PSM (Abdulaal et al., 2021), WADI (Ahmed et al., 2017) - and three cloud services datasets from our affiliation[1], i.e., Cloud-B, Cloud-S, and Cloud-Y. For comparison, we select fourteen TSAD baselines, including three traditional methods - Matrix Profile (MP) Yeh et al. (2016), KNN and KMeans, nine reconstructed-based methods - DeepSVDD (Ruff et al., 2018), DeepSAD (Ruff et al., 2019), ALOCC (Sabokrou et al., 2020), DROCC (Goyal et al., 2020), USAD (Audibert et al., 2020), DAGMM (Zong et al., 2018), AnomalyTransformer (Xu et al., 2022), TimesNet (Wu et al., 2023) and DualTF (Nam et al., 2024) and two density-based methods - GANF (Dai & Chen, 2022) and MTGFlow (Zhou et al., 2023b). The details of the datasets and baselines are shown in Appendix C and D, respectively. We implement CaPulse and baselines with PyTorch 1.10.2 on one NVIDIA A100. We follow the setting of previous works (Dai & Chen, 2022; Zhou et al., 2023b) to split datasets by 60% for training, 20% for validation, and 20% for testing. More implementation details are presented in Appendix E.

[1]For the anonymity, we have omitted the affiliation name here but will include it after paper notification.

Table 1: Comparison of 5-run AUROC, presented as the mean values with the corresponding standard deviation. The **best** / second-best results are highlighted. Ano.Trans.: AnomalyTransformer.

| | Cloud-B | Cloud-S | Cloud-Y | WADI | PSM | SMD | MSL |
|---|---|---|---|---|---|---|---|
| **MP** | 0.729 | 0.696 | 0.625 | 0.677 | 0.634 | 0.866 | 0.439 |
| **KNN** | 0.495 | 0.681 | 0.539 | 0.815 | 0.654 | 0.496 | 0.562 |
| **KMeans** | 0.565 | 0.634 | 0.709 | 0.639 | 0.535 | 0.692 | 0.521 |
| **DeepSVDD** | $0.891_{\pm0.006}$ | $0.637_{\pm0.085}$ | $0.483_{\pm0.064}$ | $0.742_{\pm0.013}$ | $0.640_{\pm0.069}$ | $0.805_{\pm0.048}$ | $0.571_{\pm0.028}$ |
| **ALOCC** | $0.725_{\pm0.120}$ | $0.716_{\pm0.120}$ | $0.587_{\pm0.030}$ | $0.709_{\pm0.080}$ | $0.651_{\pm0.120}$ | $0.712_{\pm0.060}$ | $0.504_{\pm0.016}$ |
| **DROCC** | $0.807_{\pm0.080}$ | $0.732_{\pm0.06}$ | $0.664_{\pm0.110}$ | $0.710_{\pm0.090}$ | $0.711_{\pm0.180}$ | $0.704_{\pm0.080}$ | $0.529_{\pm0.069}$ |
| **DeepSAD** | $0.867_{\pm0.027}$ | $0.642_{\pm0.079}$ | $0.453_{\pm0.056}$ | $0.723_{\pm0.009}$ | $0.644_{\pm0.076}$ | $0.818_{\pm0.055}$ | $0.521_{\pm0.011}$ |
| **DAGMM** | $0.775_{\pm0.040}$ | $0.707_{\pm0.020}$ | $0.660_{\pm0.080}$ | $0.749_{\pm0.050}$ | $0.633_{\pm0.129}$ | $0.837_{\pm0.030}$ | $0.516_{\pm0.024}$ |
| **USAD** | $0.844_{\pm0.076}$ | $0.532_{\pm0.090}$ | $0.506_{\pm0.056}$ | $0.781_{\pm0.030}$ | $0.704_{\pm0.019}$ | $0.782_{\pm0.023}$ | $0.562_{\pm0.001}$ |
| **Ano.Trans.** | $0.871_{\pm0.009}$ | $0.783_{\pm0.048}$ | $0.672_{\pm0.082}$ | $0.763_{\pm0.006}$ | $0.708_{\pm0.043}$ | $0.835_{\pm0.054}$ | $0.564_{\pm0.003}$ |
| **TimesNet** | $0.893_{\pm0.009}$ | $0.836_{\pm0.006}$ | $0.727_{\pm0.016}$ | $0.756_{\pm0.013}$ | $0.743_{\pm0.029}$ | $0.882_{\pm0.010}$ | $0.562_{\pm0.002}$ |
| **DualTF** | $0.708_{\pm0.116}$ | $0.706_{\pm0.141}$ | $0.677_{\pm0.111}$ | $0.796_{\pm0.030}$ | $0.727_{\pm0.071}$ | $0.796_{\pm0.101}$ | $0.565_{\pm0.003}$ |
| **GANF** | $0.857_{\pm0.024}$ | $0.805_{\pm0.038}$ | **$0.743_{\pm0.056}$** | **$0.843_{\pm0.005}$** | $0.725_{\pm0.010}$ | $0.772_{\pm0.055}$ | $0.443_{\pm0.037}$ |
| **MTGFLOW** | $0.884_{\pm0.013}$ | $0.842_{\pm0.028}$ | $0.728_{\pm0.044}$ | $0.822_{\pm0.018}$ | $0.721_{\pm0.035}$ | $0.836_{\pm0.023}$ | $0.570_{\pm0.003}$ |
| **CaPulse (Ours)** | **$0.926_{\pm0.007}$** | **$0.887_{\pm0.021}$** | $0.741_{\pm0.030}$ | $0.830_{\pm0.029}$ | **$0.753_{\pm0.042}$** | **$0.901_{\pm0.009}$** | **$0.604_{\pm0.017}$** |

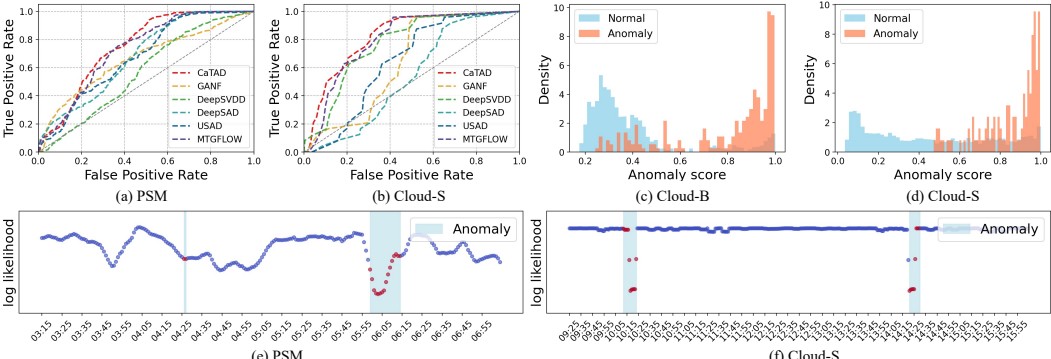

Figure 5: (a) and (b) are comparisons of AUROC curves for various models on the PSM and Cloud-S datasets, respectively. (c) and (d) are the density plots of anomaly scores for normal and anomalous instances in the Cloud-B and Cloud-S datasets. (e) and (f) visualize the log-likelihood in PSM and Cloud-S datasets.

## 5.2 EMPIRICAL RESULTS

**Model Comparison.** We follow previous density-based methods (Dai & Chen, 2022; Xu et al., 2023) to evaluate models using the Area Under the Receiver Operating Characteristic (AUROC), where higher values indicate better performance. *Quantitatively*, Table 1 reports the mean and standard deviation (STD) of AUROC scores over 5-run experiments. From these results, we can observe: 1) CaPulse achieves the highest AUROC on five out of seven datasets and ranks second on the remaining two, highlighting its robustness and consistency across various datasets. 2) CaPulse exhibits low variance, reflected by its small STD values, outperforming most baselines and demonstrating its generalizability. 3) While other NFs-based models (MTGFlow and GANF) perform well on specific datasets, they generally fall short of CaPulse, especially in cloud systems where the underlying causality of anomaly is crucial. *Graphically*, Figure 5a and 5b present the AUROC curves for two datasets, which illustrate the trade-off between the True Positive Rate (TPR) and False Positive Rate (FPR) across different threshold settings. The results show that CaPulse outperforms the baseline models by achieving higher TPRs at lower FPRs. This advantage is especially clear in low FPR regions, confirming CaPulse's reliability in minimizing false positives while detecting anomalies.

**Anomaly Score Distributions.** We first provide anomaly score distributions of the proposed model on two datasets in Figures 5c and 5d. Blue bars represent normal data, while red bars indicate anomalies. Anomalies cluster toward the higher end of the score range, near 1. For Cloud-B, normal points are spread between 0.2 and 0.6, while anomalies concentrate around 0.9 and above. In Cloud-S, the separation is more pronounced, with most anomalies scoring above 0.8, demonstrating the model's ability to effectively distinguish between normal and anomalous data.

**Log-Likelihood.** The log-likelihood behavior during anomalies of two datasets are shown in Figure 5e and 5f, respectively, where the shaded areas represent true anomalies. According to the figures, in PSM, log-likelihood drops sharply at the anomaly around 06:05, indicating the model's lower confidence during abnormal events. Similarly, in Cloud-S, the log-likelihood decreases significantly at around 10:15 and 14:25, correctly aligning with the true anomaly. These results confirm the model's effectiveness in detecting anomalies by observing clear drops in likelihood during anomalous intervals.

## 5.3 INTERPRETABILITY ANALYSIS

**True Causal Factor Identification.** Figure 6a presents the time series data, ground truth anomalies, and anomaly scores predicted by CaPulse, USAD, and MTGFlows on the Cloud-S dataset. The first four rows show different metrics changing over time and the red lines represent the anomaly labels. Time span A is a period of normal operation, while Time span B highlights abnormal events occurring in the instance (i.e., virtual machine) in the cloud computing platform. In Time span A, while there is a rise in slow tasks at around 03:52, other metrics such as CPU usage and system load remain stable, suggesting *user misoperation* might be a possible cause for it rather than a true anomaly. CaPulse captures these underlying causal factors, demonstrating its ability to detect non-obvious anomalies, while USAD does not and assigns a higher anomaly score. In contrast, during Time span B, subtle anomalies occur despite no visible abrupt changes. CaPulse captures these underlying causal factors, demonstrating its robustness in detecting non-obvious anomalies.

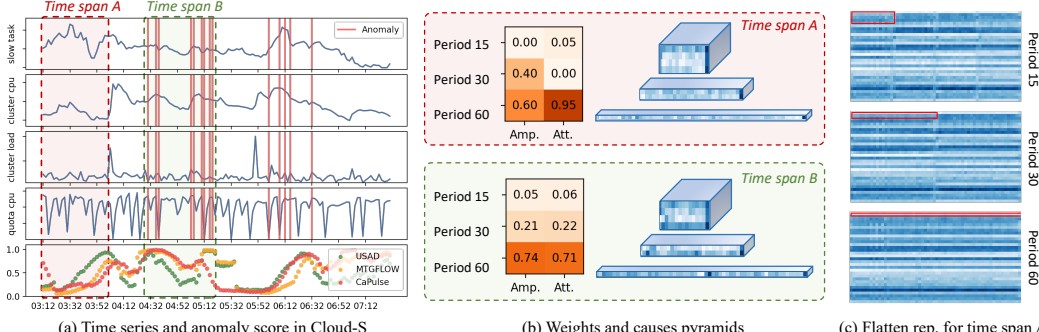

Figure 6: (a) Time series data with ground truth anomaly (*first four rows*) and predicted anomaly scores from CaPulse and other methods (*last row*). (b) Learned cause pyramids for Time spans A and B, with corresponding amplitude and attention weights. Amp.: Amplitude. Att.: Attention. (c) Flattened representations for different periods within Time span A.

Although USAD and MTGFlows also recognize this anomaly, they continue assigning high scores for 20 minutes after Time span B, failing to recognize the return to normal operation.

**Significance of Attention Mechanism.** The elevated anomaly scores predicted by CaPulse (bottom row) during Time span B align with the ground truth. Figure 6b further illustrates the causal feature pyramids, showing how feature weights differ between the two time spans. Figure 6c provides a visualization of the flattened representations to help understand the structure of the pyramids. When fusing causal factors across different periods, amplitude weights alone cannot effectively prioritize critical periods for identifying anomalies, whereas attention scores provide this capability. As shown in Figure 6b, during Time span A, although the amplitude weights assign similar importance to Periods 30 and 60, the high attention score for Period 60 (0.95) highlights that long-term features are more relevant for capturing causal factors. This is particularly important when addressing short-term "user misoperations", where focusing only on short-term patterns could result in misinterpretations. The attention mechanism mitigates this risk by directing focus to the most relevant periods.

**Interpretability of Causal Representations.** Next, we analyze the interpretability of the "causal rhythm" learned by CaPulse, i.e., the representation of latent causal factors $C_{\text{ind}}$. The analysis uses the Cloud-S dataset, with 10 latent causal factors ($N = 10$) denoted as $\{c_1, c_2, \ldots, c_{10}\}$. We then present an interpretability analysis using the SHapley Additive exPlanations (SHAP) (Lundberg & Lee, 2017) on these causal representations. In essence, SHAP helps explain how each latent cause contributes to the anomaly detection labels. The interpretability results are visualized in Figure 7. Red (positive SHAP values) indicates a push towards anomaly detection, while blue (negative SHAP values) indicates a shift towards normal behavior. According to the result, we have the following observations: 1) The waterfall plot in Figure 7a presents the contribution of each cause for a specific sample, where $c_1$ contributes the most positively, pushing the prediction towards the anomaly

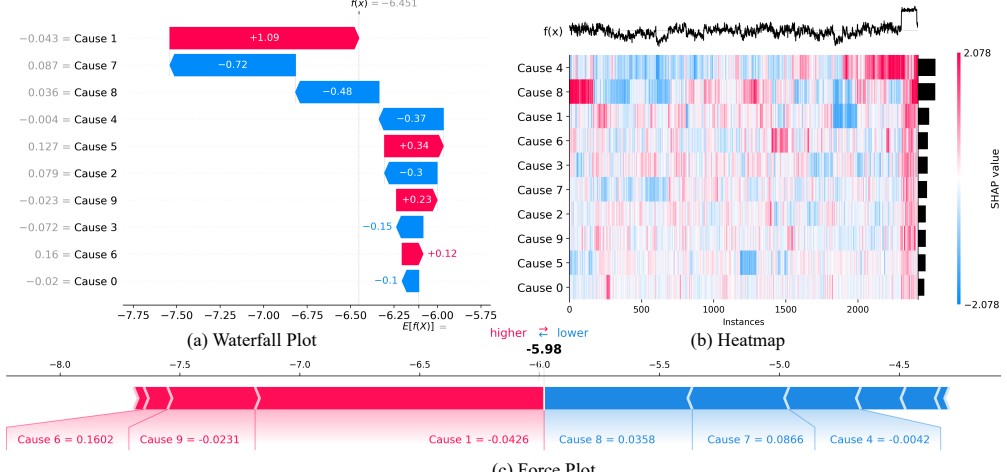

Figure 7: (a) Waterfall plot: SHAP values for an individual prediction showing contributions from each cause. (b) Heatmap: SHAP values across multiple instances and causes. (c) Force plot: individual feature contributions for a specific instance. Cause $i$ refers to the $i$-th latent causal factor $c_i$.

exist, yet $c_7$ has the most significant negative impact, shifting the prediction towards normal behavior. 2) The heatmap in Figure 7b provides a global overview of how the causes impact identifying anomalies across multiple samples. Each row represents a latent cause, and each column represents a sample. $c_1$, $c_4$ and $c_8$ show consistently high positive SHAP values for many instances, while $c_5$ and $c_7$ stand out with significant negative SHAP values across many instances. 3) The force plot in Figure 7c provides a detailed view of how these causes push or pull a specific detection from the average value to the final prediction. In this sample, $c_6$ drives the prediction towards anomaly, while $c_9$ highly recognizes the sample is normal. $c_1$ and $c_7$ show moderate contributions.

In summary, causes like $c_1$ how consistently demonstrate a strong positive influence on anomaly detection, indicating that its representation is closely linked to anomaly-indicating patterns (e.g., "hardware failure" in a cloud service context). Conversely, causes like $c_7$ tend to shift predictions toward normal behavior, suggesting that these causes are more reflective of regular instances (e.g., "users' misperception"). Detailed experimental settings and plot explanations are provided in Appendix B.2.

### 5.4 ABLATION STUDY & HYPERPARAMETER SENSITIVITY

**Effects of Core Components.** To evaluate the contribution of each core component in Ca-Pulse, we conducted an ablation study using the following variants: a) **w/o CI**, which removes causal intervention and the similarity loss; b) **w/o ICM**, which excludes the ICM principle, thereby not ensuring joint independence of causal factors; c) **w/o Attn**, which omits the attention mechanism used for fusing

Table 2: Variant results on two datasets.

| Variant | SMD | Cloud-S |
|---------|-----|---------|
| **w/o CI** | 0.890±0.015 (↓1.87%) | 0.825±0.056 (↓6.99%) |
| **w/o ICM** | 0.884±0.010 (↓2.54%) | 0.848±0.005 (↓4.40%) |
| **w/o Attn** | 0.888±0.012 (↓2.09%) | 0.859±0.016 (↓3.16%) |
| **w/o GP** | 0.889±0.015 (↓1.98%) | 0.856±0.011 (↓3.49%) |
| **CaPulse** | **0.901**±0.009 | **0.887**±0.021 |

multi-period features; and d) **w/o GP**, which excludes global period information by not applying the PC-Mask in the normalizing flows. Table 2 reports their AUROC results across two datasets, showing that all components contribute significantly to the model's overall performance. Notably, for Cloud-S, excluding causality-related components (**w/o CI** and **w/o ICM**) results in a marked performance degradation, underscoring the importance of causal mechanisms in cloud environments. More ablation results are presented in Appendix F.1.

**Hyperparameter Sensitivity.** Figure 8 illustrates the impact of different configurations of # Layers, # Blocks, and the balance coefficients in the loss function, $\alpha$ and $\beta$, on the model's AUROC performance for the SMD dataset. In Figure 8a, increasing the number of blocks consistently improves performance, while the number of layers has a lesser effect, with the best AUROC achieved at 2 layers and 5 blocks. Figure 8b reveals the sensitivity to $\alpha$ and $\beta$, showing optimal AUROC when both parameters are set around 0.01. This underscores the need to balance the contributions of different loss terms for optimal performance and stability.

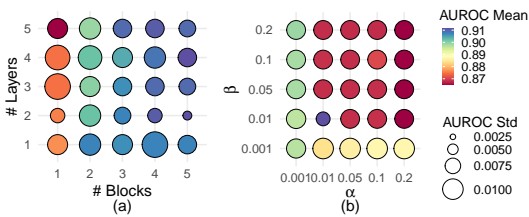

Figure 8: Study on hyperparameter combinations on AUROC for the SMD dataset.

## 6 CONCLUSION & DISCUSSION

In this paper, we present the first attempt to take a causal perspective for TSAD and implement it within a deep learning framework. Concretely, building on the proposed SCM, we introduce Ca-Pulse, a causality-driven deep learning model designed to detect anomalies by leveraging causal tools while addressing key challenges in TSAD, including label scarcity, data imbalance, and multiple periodicities. Extensive experiments on seven datasets across five domains demonstrate CaPulse is equipped to effectively detect both subtle and significant deviations, enhancing interpretability and robustness. A potential limitation of CaPulse is its reliance on the assumption that anomalies lie in low-density regions, which may not always hold in complex real-world scenarios — for instance, in high-frequency trading data where significant anomalies may cluster in high-density regions during market events or crashes. Future work could explore relaxing these distributional assumptions and incorporating reversible transformations to generate synthetic anomalies.

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

## A  NORMALIZING FLOWS FOR TIME SERIES

**Normalizing Flows.** Normalizing Flows (NFs) (Tabak & Turner, 2013; Papamakarios et al., 2021) are a powerful technique for density estimation, widely utilized in tasks such as image generation (Papamakarios et al., 2017). Essentially, NFs are invertible transformations that map data from an input space $\mathbb{R}^D$ to a latent space $\mathbb{R}^D$, such that a complex distribution $P_{\mathcal{X}}$ on the input space $\boldsymbol{X} \in \mathbb{R}^D$ is transformed into a simpler distribution $P_{\mathcal{Z}}$ in the latent space $\boldsymbol{Z} \in \mathbb{R}^D$. These mappings, $\mathcal{F} \colon \mathcal{X} \mapsto \mathcal{Z}$, are typically constructed as a series of invertible functions. By utilizing the change of variables formula, the probability density function $P_{\mathcal{X}}(\boldsymbol{X})$ is expressed as:

$$P_{\mathcal{X}}(\mathbf{X}) = P_{\mathcal{Z}}(\mathbf{Z}) \left| \det\left( \frac{\partial \mathcal{F}(\mathbf{X})}{\partial \mathbf{X}} \right) \right|, \tag{7}$$

where $\frac{\partial \mathcal{F}(\boldsymbol{X})}{\partial \boldsymbol{X}}$ is the Jacobian matrix of the transformation $\mathcal{F}$ at $\boldsymbol{X}$. NFs offer two key advantages: both the inverse transformation $\boldsymbol{X} = \mathcal{F}^{-1}(\boldsymbol{Z})$ and the computation of the Jacobian determinant can be efficiently computed, with the determinant calculation typically taking $O(D)$ time. This enables the following expression for the log-likelihood of the data under the flow:

$$\log P_{\mathcal{X}}(\mathbf{X}) = \log P_{\mathcal{Z}}(\mathbf{Z}) + \log |\det(\partial \mathbf{Z}/\partial \mathbf{X})|. \tag{8}$$

**Temporal Conditioned Normalizing Flows.** To adapt NFs for time series data, temporal conditioned flows introduce additional conditional information, denoted as $\mathbf{h} \in \mathbb{R}^{D_h}$, which may have a different dimension from the input. The flow is now expressed as $\mathcal{F} \colon \mathbb{R}^D \times \mathbb{R}^{D_h} \to \mathbb{R}^D$, allowing for conditioning on temporal features. The log-likelihood of the time series $\mathbf{X}$, conditioned on the temporal context $\mathbf{h}$, is given by:

$$\log P_{\mathcal{X}}(\mathbf{X}|\mathbf{h}) = \log P_{\mathcal{Z}}(\mathcal{F}(\mathbf{X};\mathbf{h})) + \log |\det(\nabla_{\mathbf{X}} \mathcal{F}(\mathbf{X};\mathbf{h}))|. \tag{9}$$

**Coupling Layers.** One of the key innovations in NFs proposed by a widely-used variant Real-NVP (Dinh et al., 2017) is the use of *coupling layers*, which simplify the computation of the Jacobian determinant. In a coupling layer, part of the input remains unchanged, while another part is transformed. Specifically, the transformation is defined as:

$$\begin{cases} \boldsymbol{Y}^{1:d} = \boldsymbol{X}^{1:d}, \\ \boldsymbol{Y}^{d+1:D} = \boldsymbol{X}^{d+1:D} \odot \exp(\mathcal{S}_{\theta}(\boldsymbol{X}^{1:d})) + \mathcal{T}_{\theta}(\boldsymbol{X}^{1:d}), \end{cases} \tag{10}$$

where $\odot$ represents element-wise multiplication, $\mathcal{S}(\cdot)$ is a scaling function, and $\mathcal{T}(\cdot)$ is a translation function, both parameterized by $\theta$. The coupling layer thus enables efficient transformations by only modifying part of the input at a time. To achieve complex, nonlinear density mappings, multiple coupling layers are stacked, alternating which dimensions are transformed at each layer. This ensures that all dimensions are transformed over the course of the flow, while keeping computations efficient.

## B  DETAILS OF CAPULSE

### B.1  ARCHITECTURE OF PACM & MPCF

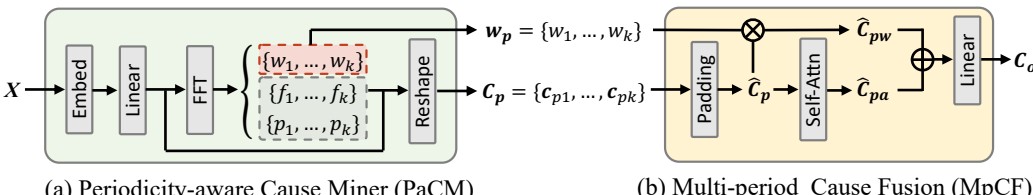

(a) Periodicity-aware Cause Miner (PaCM)  (b) Multi-period  Cause Fusion (MpCF)

Figure 9: Architecture of the proposed PaCM and MpCF modules.

We design PaCM and MpCF to handle multiple local periodicities in time series data by extracting and fusing periodic information at various levels, illustrated in Figure 9a and 9b, respectively. Inspired by TimesNet (Wu et al., 2023), PaCM is responsible for capturing different period levels of information from the input time series $\boldsymbol{X}$. PaCM first embeds the input time series $\boldsymbol{X}$, followed by a linear transformation to project the input into a higher-dimensional space. Next, an FFT is applied to obtain the frequency components $\{f_1, f_2, \ldots, f_k\}$ and their corresponding periodicities $\{p_1, p_2, \ldots, p_k\}$ and the amplitude weight $\boldsymbol{w}_p = \{w_1, w_2, \ldots, w_k\}$. The output of PaCM is a set of reshaped latent causal factors $\mathbf{C}_p = \{\boldsymbol{C}_{p1}, \boldsymbol{C}_{p2}, \ldots, \boldsymbol{C}_{pk}\}$, where each $\mathbf{C}_{pi}$ represents a representation for the $i$-th period to consist the pyramid $\mathbf{C}_p \in \mathbb{R}^{N \times D_h \times k}$.

MpCF is designed to fuse the multi-period information extracted by PaCM. MpCF begins by padding the causal factors from different periods, followed by applying a self-attention mechanism to compute attention scores for each period. These attention scores, along with the amplitude weights $\boldsymbol{w}_p$, are used to weight the periodic components and produce the final omni-causal representation $\hat{\boldsymbol{C}}_p$. The final output of MpCF is the fused causal representation $\boldsymbol{C}_o \in \mathbb{R}^{N \times D_h}$, which integrates the most relevant information from all periods. The advantages of the attention mechanism are demonstrated by the improvement of performance (see Section 5.3) and interpretability (see Section 5.4). Together, PaCM and MpCF effectively capture and fuse periodic information, enabling the model to handle complex, multi-periodic time series data.

## B.2 CAUSAL REPRESENTATION ANALYSIS

In Section 5.3, we analyze the interpretability of the "causal rhythm" learned by the proposed model. Here we provide details on the analysis experiment. The experiment was conducted on the Cloud-S dataset, with the number of latent causal factors set to 10, thus resulting in 10 distinct learned causal representations. We present an interpretability analysis using SHAP (Lundberg & Lee, 2017), SHAP helps explain how each latent cause contributes to the anomalies. Specifically, we first train an XGBoost classifier (Chen & Guestrin, 2016) using the learned causal representations to predict the anomaly labels. The SHAP values derived from this model quantify the contribution of each cause to the prediction—indicating how much each cause increases or decreases the likelihood of an anomaly—thereby providing interpretability to the learned representations. For clarity in the analysis, we refer to the latent causes as $\boldsymbol{c}_1$ through $\boldsymbol{c}_{10}$, and the following 'model' is the XGBoost instead of CaPulse. The results are visualized in three SHAP plots (Figure 7), each offering unique insights into how individual or grouped causes influence the model's predictions. We have already presented the observation in the main text, so here we just give some explanation about these SHAP plots as follows:

- The waterfall plot presents the contribution of each cause for a specific instance (one sample). Starting from the average output of the XGBoost model, the contribution of each cause pushes the prediction either towards predicting an anomaly (in red) or towards predicting normal behavior (in blue).

- The heatmap provides a global overview of how the causes impact predictions across multiple instances. Each row represents a learned cause, and each column represents an instance from the dataset. The color intensity indicates the SHAP value, with red representing a positive contribution towards predicting anomalies and blue representing a negative contribution towards normal behavior.

- The force plot provides a detailed view of how causes push or pull a specific prediction from the base value to the final predicted score. Red arrows represent causes that increase the predicted score (i.e., lead towards an anomaly), while blue arrows represent causes that decrease the score (i.e., lead towards normal behavior).

## B.3 COMPUTATIONAL COMPLEXITY

For simplicity, we omit hidden dimensionality in the following analysis. Given that $T$ denotes the number of time steps, the computational complexity of the FFT process is $\mathcal{O}(T \log T)$, which is performed in obtaining the global and the local periods. The first stage, i.e., getting the global period is a preprocessing step for the dataset and, thus is not included in the training process. The second stage, i.e., getting the local period occurs within the PaCM module. Additionally, the attention

mechanism in the MpCF module introduces a complexity of $\mathcal{O}(N^2 D_h)$, where $N$ indicates the number of causal factors and $D_h$ describes the hidden dimensionality. The transformations in the PeNF are linear. Thus the total complexity is $\mathcal{O}(T \log T) + \mathcal{O}(N^2 D_h)$.

## C  DETAILS OF DATASETS

We evaluate the proposed model on seven real-world datasets from different domains, including five commonly used public datasets for TSAD - MSL (Mars Science Laboratory rover) (Hundman et al., 2018), SMD (Server Machine Dataset) (Su et al., 2019), PSM (Pooled Server Metrics) (Abdulaal et al., 2021), WADI (Water Distribution) (Ahmed et al., 2017) - and three cloud computing platform datasets, namely Cloud-B, Cloud-S, and Cloud-Y, collected by our company [2]. Each dataset consists of multivariate monitoring metrics recorded at different time points from a single instance (i.e., virtual machine). These metrics include factors such as the number of slow-running tasks, CPU usage, and memory consumption. The labels indicate whether any issues occurred in the monitored instance.

Table 3: Detail of datasets. # Train/Val/Test: the number of training/validation/test samples.

| Dataset | # Dims | # Train | # Val | # Test | Anomaly Rate (%) |
|---------|--------|---------|-------|--------|------------------|
| **Cloud-B** | 6 | 14,604 | 4,868 | 4,869 | 5.649 |
| **Cloud-S** | 6 | 14,604 | 4,868 | 4,869 | 4.453 |
| **Cloud-Y** | 6 | 14,604 | 4,868 | 4,869 | 2.703 |
| **WADI** | 123 | 103,680 | 34,560 | 34,561 | 5.774 |
| **PSM** | 25 | 52,704 | 17,568 | 17,569 | 27.756 |
| **SMD** | 38 | 14,224 | 4,741 | 4,742 | 3.037 |
| **MSL** | 55 | 44,237 | 14,745 | 14,746 | 10.533 |

## D  DETAILS OF BASELINES

We opted to include a selection of widely-used cutting-edge methods for comparative evaluation. We describe these baselines used in our experiments and their settings as follows. We use the same setting for all datasets.

- **DeepSVDD** (Ruff et al., 2018) Deep Support Vector Data Description (DeepSVDD) is a deep learning-based anomaly detection method that minimizes the volume of a hypersphere enclosing the normal data in the latent space. We utilize the publicly available implementation[3] for our experiments. The hidden dimension is set to 64, the number of layers are set to 2.

- **ALOCC** (Sabokrou et al., 2020): Adversarially Learned One-Class Classifier (ALOCC) leverages GANs to learn compact representations of normal data for detecting anomalies. We use the official implementation[4] provided by the authors. We set the hidden dimension to 64 and the number of layers to 2.

- **DROCC** (Goyal et al., 2020): Deep Robust One-Class Classification (DROCC) is a method that generates adversarial perturbations around the normal data to improve robustness for anomaly detection. The authors' code[5] is employed for our experiments. The model uses a hidden dimension of 64 and consists of 2 layers. We set gamma (parameter to vary projection) to 2 and lamda (weight given to the adversarial loss) to 0.0001.

- **DeepSAD** (Ruff et al., 2019): Deep Semi-Supervised Anomaly Detection (DeepSAD) builds on DeepSVDD by incorporating labeled anomalies during training, aiming for improved detection of

---

[2]Company details temporally omitted for anonymity.

[3]https://github.com/lukasruff/Deep-SVDD-PyTorch

[4]https://github.com/khalooei/ALOCC-CVPR2018

[5]https://github.com/microsoft/EdgeML/tree/master/pytorch

rare anomalies. We adopt the publicly released code[6] for our analysis. A hidden dimension of 64 is employed, with the number of layers fixed at 2.

- **DAGMM** (Zong et al., 2018): Deep Autoencoding Gaussian Mixture Model (DAGMM) jointly optimizes a deep autoencoder and a Gaussian mixture model to detect anomalies based on reconstruction errors and low-dimensional latent representations. We leverage the code[7] shared by the authors. The hidden size is defined as 64, and the network is composed of 2 layers.

- **USAD** (Audibert et al., 2020): UnSupervised Anomaly Detection (USAD) is an unsupervised method designed for multivariate time series, using autoencoders to learn normal patterns and detect anomalies. The authors' implementation[8] is employed in our study. For this configuration, the hidden dimension is 64, and the model has 2 layers. $\alpha$ and $\beta$ are both set to 0.5.

- **AnomalyTransformer** (Xu et al., 2022): Anomaly Transformer introduces a novel approach for unsupervised time series anomaly detection by leveraging an Association Discrepancy criterion, an innovative Anomaly-Attention mechanism, and a minimax strategy to enhance the differentiation between normal and abnormal patterns. The official code[9] is employed for our experiments. The window size is set to 60, the number of attention heads is 8, and the feedforward network dimension is 512.

- **GANF** (Dai & Chen, 2022): Graph-Augmented Normalizing Flows (GANF) leverages normalizing flows conditioned on a graph neural network for unsupervised anomaly detection in multivariate time series. We utilize the official code[10] for our experiments. We configure the hidden size to 32 and set the number of blocks to 1.

- **MTGFlow** (Zhou et al., 2023b): MTGFlow uses entity-aware normalizing flows to capture multiscale dependencies in time series data for anomaly detection. We rely on the authors' released code[11] for conducting our experiments. The setup involves a hidden dimension of 32 and a total of 2 layers.

# E  EXPERIMENT SETTINGS

Our model is trained using Adam optimizer (Kingma & Ba, 2014) with a learning rate of 0.001. We implement the high-frequency threshold $k_h = 25\%T$ in causal intervention in Eq. 1 and the amplitude of intervention $\sigma$ we search over $\{0.01, 0.1, 1, 2, 10\}$. For the hidden dimension $D_h$, we conduct a grid search over $\{8, 16, 32, 64\}$. For the number of layers and blocks, we test it from 1 to 5. The balance coefficients in the loss function $\alpha$ and $\beta$ are searched over $\{0.001, 0.1, 0.05, 0.1, 0.2\}$. We outline the optimal hyperparameter configurations used for CaPulse across different datasets:

- **Cloud-B:** We set the hidden size to 32, the number of blocks to 2, and the number of layers to 2. The balancing coefficients for the mutual information loss, $\alpha$, and $\beta$, were both set to 0.1, ensuring an appropriate trade-off between different loss components.

- **Cloud-S:** For Cloud-S, the hidden size is set to 32, with 2 blocks and 1 layer. The mutual information loss coefficients $\alpha$ and $\beta$ were set to 0.01 and 0.1, respectively.

- **Cloud-Y:** In this case, the hidden size was set to 32, the number of blocks to 3, and the number of layers to 1. The mutual information loss coefficients $\alpha$ and $\beta$ were both set to 0.1.

- **WADI:** The WADI dataset used a hidden size of 32, with 1 block and 1 layer. The mutual information loss coefficients $\alpha$ and $\beta$ were both set to 0.05.

- **PSM:** For PSM, we configured the model with a hidden size of 32, 1 block, and 1 layer. The mutual information loss coefficients were set to $\alpha = 0.1$ and $\beta = 0.1$.

- **SMD:** The model for SMD was also configured with a hidden size of 32, 5 blocks, and 2 layers. The balancing coefficients for the mutual information loss were both set to 0.01.

---

[6]https://github.com/lukasruff/Deep-SAD-PyTorch

[7]https://github.com/danieltan07/dagmm

[8]https://github.com/manigalati/usad

[9]https://github.com/thuml/Anomaly-Transformer

[10]https://github.com/EnyanDai/GANF

[11]https://github.com/zqhang/MTGFLOW

- **MSL:** For the MSL dataset, we set the hidden size to 32, the number of blocks to 1, and the number of layers to 1. The mutual information loss coefficients $\alpha$ and $\beta$ were both set to 0.1.

# F  MORE EXPERIMENTAL RESULTS

## F.1  ABLATION STUDIES

To further demonstrate the generalizability of our approach, we conducted ablation studies on two additional datasets beyond those described in Section 5.4. The results of these experiments are presented in Table 4. The results show that removing any single component leads to noticeable performance drops, ranging from 3.46% to 4.1% on Cloud-B and 3.59% to 3.98% on PSM. In contrast, the full CaPulse model consistently achieves the highest performance.

Table 4: Variant results on the Cloud-B and PSM datasets.

| Dataset | Cloud-B | PSM |
|---|---|---|
| **w/o CI** | $0.888 \pm 0.002$ ($\downarrow$4.1%) | $0.726 \pm 0.009$ ($\downarrow$3.59%) |
| **w/o ICM** | $0.889 \pm 0.006$ ($\downarrow$4%) | $0.725 \pm 0.002$ ($\downarrow$3.72%) |
| **w/o Attn** | $0.891 \pm 0.002$ ($\downarrow$3.78%) | $0.723 \pm 0.01$ ($\downarrow$3.98%) |
| **w/o GP** | $0.894 \pm 0.001$ ($\downarrow$3.46%) | $0.725 \pm 0.009$ ($\downarrow$3.72%) |
| **CaPulse** | $0.926 \pm 0.007$ | $0.753 \pm 0.042$ |

## F.2  EFFICIENCY COMPARISON

We compare our method with some classical baselines to demonstrate the methods' efficiency. For theoretical computational complexity, we have discussed in Appendix B.3.

**Time Cost and Parameter Comparison.** We first compare the time cost and parameter of CaPulse and one of the classical TSAD method Matrix Profile (MP) (Yeh et al., 2016). Theoretically, the complexity of MP is $\mathcal{O}(T_l^2 \log T_l)$, where $T_l$ represents the total length of the time series (typically, $T_l \gg T$). Thus, MP's theoretical complexity is higher than that of our approach. We conducted experiments on four datasets and measured the time costs. Note, that we believe that a direct efficiency comparison may be unfair for several reasons: 1) Methods like MP can only be run on the CPU, while DL methods such as CaPulse can leverage GPU acceleration. 2) MP operates directly on the test data, which is smaller (about one-third of the training set size), whereas CaPulse is trained on the full training set. 3) Training epochs vary across datasets and can be adjusted, making the total training time flexible. Thus, to provide additional context, we also included a comparison with a recent DL-based method, DualTF Nam et al. (2024). The results are summarized in Table 5, where we observe that CaPulse achieves significantly lower time costs per epoch and consistently outperforms both MP and DualTF in ROC scores, demonstrating both efficiency and effectiveness.

**Additional classical baselines.** To further compare our method with classical baselines, we have compared it with three additional baselines, i.e., MP, KNN, and K-means. The results are shown in Table 6, which demonstrate that CaPulse consistently achieves superior ROC scores compared to classical methods, reinforcing its robustness and accuracy in detecting anomalies across diverse datasets.

# G  MORE DISCUSSIONS

## G.1  APPLICABILITY OF THE PROPOSED SCM IN REAL-WORLD SCENARIOS

In Section 3.1, we introduced a causal perspective on the TSAD task by proposing a Structural Causal Model (SCM), as illustrated in Figure 2b. In the proposed SCM, the non-causal factors $U$ and the causal factors $C$ are defined as unobserved latent variables that represent a range of potential influences. Based on whether a factor directly causes $y$ or only affects $X$ without impacting $y$, we can categorize it as either a causal factor $C$ or a non-causal factor $U$. This distinction is therefore flexible and may vary depending on the specific domain or scenario. We acknowledge that real-world

Table 5: Comparison of efficiency of methods across datasets. The magnitude of #Param (the number of parameters) is Kilo. Time is reported in seconds for MP and seconds per epoch for DualTF and CaPulse.

| Dataset | Metric | MP | DualTF | CaPulse |
|---------|--------|-----|--------|---------|
| PSM | #Param (K) | - | 4801.6 | 204.7 |
|  | Time | 25.944 | 2.265 ± 0.356 | 0.533 ± 0.192 |
|  | ROC-AUC | 0.634 | 0.727 ± 0.071 | 0.753 ± 0.042 |
| SMD | #Param (K) | - | 4820 | 264.7 |
|  | Time | 24.673 | 0.709 ± 0.385 | 0.182 ± 0.195 |
|  | ROC-AUC | 0.866 | 0.796 ± 0.101 | 0.906 ± 0.009 |
| WADI | #Param (K) | - | 4949.1 | 342.2 |
|  | Time | 40.428 | 4.52 ± 0.372 | 2.505 ± 0.197 |
|  | ROC-AUC | 0.677 | 0.796 ± 0.030 | 0.830 ± 0.029 |
| SWaT | #Param (K) | - | 4840.5 | 242.4 |
|  | Time | 43.065 | 11.244 ± 0.34 | 3.613 ± 0.243 |
|  | ROC-AUC | 0.600 | 0.769 ± 0.019 | 0.782 ± 0.004 |

Table 6: Comparison with classical baselines.

|  | SWaT | WADI | PSM | SMD |
|---|------|------|-----|-----|
| MP | 0.600 | 0.677 | 0.634 | 0.866 |
| KNN | 0.716 | 0.815 | 0.654 | 0.496 |
| K-means | 0.560 | 0.639 | 0.535 | 0.692 |
| CaPulse | 0.782 ± 0.004 | 0.830 ± 0.029 | 0.753 ± 0.042 | 0.906 ± 0.009 |

environments can be more complex and dynamic than our model assumptions. Nevertheless, we believe that *fundamental* patterns in real-world settings can be effectively abstracted and represented within this SCM framework for TSAD.

To further support this point, in addition to the cloud computing platform example provided in Section 3.1, we offer another real-world scenario in healthcare. In this context, $X$ could represent biometric data (e.g., heart rate, sleep patterns) collected from wearable devices, with anomalies $y$ indicating potential health issues. Here, $U$ might correspond to environmental factors or background noise that influence the readings in $X$ without signifying genuine bodily anomalies, while $C$ could represent factors such as medication side effects that directly impact both $X$ and $y$. Thus, this adaptability enables our model to accommodate different domains by appropriately classifying factors as causal or non-causal based on their direct or indirect influence on the anomaly outcome.

## G.2 AUGMENTATION METHODS FOR CAUSAL INTERVENTION

In Section 4.1, we perform causal intervention by adding noise to the less significant part of the input data, the high-frequency components, to simulate real-world disturbances. We recognize that real-world scenarios can be more complex, so we tested multiple augmentation methods on two different datasets, PSM and SMD. The ROC results are presented in Table 7.

Here, **HighFreq** refers to adding noise to the high-frequency components, which was our initial approach. **LowFreq** denotes adding noise to the low-frequency components, and **Shift** indicates shifting the input time series by 20 time steps. The "+" symbol represents a combination of different methods. According to the results, adding noise to high-frequency components yields the best performance, with **LowFreq** also performing well but slightly below **HighFreq**. Shifting the time series has a lesser impact, and combining multiple augmentation methods does not improve performance beyond **HighFreq** alone, suggesting that excessive variability may obscure meaningful causal patterns.

Table 7: 5-run results for different augmentation methods to implement causal intervention.

| Augmentation Method | PSM | SMD |
|---|---|---|
| **HighFreq** | **0.753** ± 0.042 | **0.906** ± 0.009 |
| **LowFreq** | 0.743 ± 0.015 | 0.902 ± 0.007 |
| **Shift** | 0.728 ± 0.011 | 0.885 ± 0.022 |
| **HighFreq + LowFreq** | 0.725 ± 0.009 | 0.905 ± 0.005 |
| **HighFreq + Shift** | 0.727 ± 0.011 | 0.884 ± 0.021 |
| **LowFreq + Shift** | 0.725 ± 0.008 | 0.881 ± 0.018 |
| **HighFreq + LowFreq + Shift** | 0.729 ± 0.014 | 0.874 ± 0.010 |

## G.3 COMPARISON WITH RELATED WORKS

CaPulse addresses key gaps in existing methods for TSAD by introducing a causal and periodicity-aware approach. Density-based TSAD methods such as GANF (Dai & Chen, 2022) and MT-GFlow (Xu et al., 2023) lack a causal perspective and fail to account for multi-periodicity in time series data. While forecasting- (Hundman et al., 2018; Shen et al., 2020) and reconstruction-based models (Su et al., 2019; Audibert et al., 2020; Xu et al., 2022) improve anomaly detection, they rely solely on statistical patterns without capturing underlying causal processes, making them less robust to noise and dynamic changes. In contrast, CaPulse enhances interpretability and robustness by integrating a causal view and density estimation, specifically tailored to handle complex temporal rhythms. Additionally, causal inference-based methods like Cost (Woo et al., 2022) and CaseQ (Yang et al., 2022) focus on time series forecasting and sequential event prediction, respectively, but do not address anomaly detection. CaPulse is uniquely designed for TSAD, focusing on the generative processes behind anomalies to improve detection accuracy and provide deeper insights into the underlying causes of anomalies in time series data.

