# OpenReview forum: "CaPulse: Detecting Anomalies by Tuning in to the Causal Rhythms of Time Series"
_ICLR.cc/2025/Conference — Submitted to ICLR 2025_

### Official Review · Reviewer_kFnw · 2024-10-22

**Soundness:** 2
**Presentation:** 3
**Contribution:** 3
**Rating:** 8
**Confidence:** 5

**Summary:**

This paper proposed CaPulse, which is a new causality-based framework for time series anomaly detection. The framework includes PaCM, MpCF moduls for causal treatment and multimple periodicities. This approach is periodicity-aware and density-based anomaly detection. Unlike traditional approaches that may fail to capture the underlying mechanisms of anomaly generation, CaPulse builds a structural causal model to understand the root causes of time-series anomalies. The experiments show better accuracy and interpretability than exiting methods.

**Strengths:**

- The CaPulse framework presents a novel approach to time-series anomaly detection by introducing causal inference. In this point, this work is different from traditional deep learning-based methods.
- Another innovative aspect is the introduction of Periodic Normalizing Flows(PeNF) with a mask mechanism for periodicity awareness. This approach is particularly well-suited for time series with complex multi-periodicity, enhancing both anomaly detection performance and interpretability.
- This paper provides empirical evidence to support the claims, with interpretability analysis.

**Weaknesses:**

- From section 3, a causal view of TSAD includes hard assumptions that might not hold in various real-world settings.
- The limited number of baselines and benchmark datasets.
- There is no friendly explanation for the interpretability plot, especially in Figure 7.

**Questions:**

1. As mentioned in the paper, I wonder if some non-causal factors(U) such as “user malfunction” or “data collection jitter” can also be considered as causal factors depending on the domain?
2. What is the rationale for augmenting the raw input time series in the pipeline of CaPulse? Does the method of augmentation influence the performance of the entire framework? Or would it be more helpful to have multiple ways of augmenting instead of just one? Is it enough to simulate real-world disturbances?
3. How to determine an anomaly judgment based on an anomaly score? Is there a threshold?
4. What is the meaning of Figure 6c? What was the author trying to express?
5. In Figure 6a, I don't understand why CaPulse is the only one that can accurately predict the anomalies because I don't know why they are anomalies through the time-series plot.

---

> ### Author Response · Authors · 2024-11-16
> **Response to Weaknesses by Authors**
>
> We sincerely thank you for your valuable comments and for recognizing the novelty of our approach, the effectiveness of Periodic Normalizing Flows for periodicity handling, and our focus on interpretability. Below, we address the concerns you have raised point by point.
>
> **[Weaknesses]**
>
> **W1.** The causal view of TSAD might not hold in various real-world settings.
>
> **A1.**  Thank you for your insightful comments. We acknowledge that real-world scenarios can be more complex and dynamic. However, we believe that the *fundamental* patterns in the real world can be abstracted and represented within the proposed SCM for TSAD. To further support this point, in addition to the cloud computing platform example in **Section 3.1**, we provide another real-world scenario in healthcare: here, $\boldsymbol{X}$ could represent biometric data (e.g., heart rate, sleep patterns) from wearable devices, with anomalies $\boldsymbol{y}$ indicating potential health issues. In this context, $\boldsymbol{U}$ might represent environmental factors or background noise that can influence the reading $\boldsymbol{X}$ without indicating true bodily anomalies, while $\boldsymbol{C}$ can be factors like medication side effects, which directly impact both $\boldsymbol{X}$ and $\boldsymbol{y}$. We have added a discussion in **Appendix G.1** to clarify these points. Thank you for this valuable comment which has significantly improved our paper.
>
> **W2.** The limited number of baselines and benchmark datasets.
>
> **A1.** Thank you for your valuable feedback. To address your concern, we added five additional baselines, including three classical methods (MatrixProfile [1], KNN, and KMeans) and two recent baselines (DualTF [2] and TimesNet [3]), as well as an additional dataset (SWaT [4]). The preliminary five-run results are shown in the table below (*Italicized* entries indicate newly added experimental results). Due to the time constraint, we commit that all experiments will be conducted in full, and be included in our final revision.
> |                     | **SWaT**        | **WADI**        | **PSM**         | **SMD**         |
> |---------------------|-----------------|-----------------|-----------------|-----------------|
> | **_MatrixProfile_** | _0.600_         | _0.677_         | _0.634_         | _0.866_         |
> | **_KNN_**           | _0.716_         | _0.815_         | _0.654_         | _0.496_         |
> | **_Kmeans_**        | _0.560_         | _0.639_         | _0.535_         | _0.692_         |
> | **DAGMM**           | _0.654 ± 0.093_ | 0.749 ± 0.050   | 0.633 ± 0.129   | 0.837 ± 0.030   |
> | **USAD**            | _0.777 ± 0.019_ | 0.781 ± 0.030    | 0.704 ± 0.019   | 0.782 ± 0.023   |
> | **GANF**            | _0.788 ± 0.002_ |  0.843 ± 0.005  | 0.725 ± 0.010   | 0.772 ± 0.055   |
> | **MTGFLOW**         | _0.757 ± 0.019_ | 0.822 ± 0.018   | 0.721 ± 0.035   | 0.836 ± 0.023   |
> | **_DualTF_**        | _0.769 ± 0.019_ | _0.796 ± 0.030_ | _0.727 ± 0.071_ | _0.796 ± 0.101_ |
> | **_TimesNet_**      | **_0.789 ± 0.016_** | _0.756 ± 0.013_ | _0.743 ± 0.029_ | _0.882 ± 0.010_ |
> | **CaPulse**         | _0.782 ± 0.004_ | **0.830 ± 0.029**   | **0.753 ± 0.042**   | **0.906 ± 0.009**   |
>
>
> **W3.** Lack of friendly explanation for Figure 7.
>
> **A2.** Thank you for your comments, and we apologize for the unclear explanation of Figure 7. We have provided a more reader-friendly explanation below: In Figure 7, **red (positive SHAP values) indicates a push towards anomaly detection, while blue (negative SHAP values) indicates a shift towards normal behavior. The magnitude of each value reflects the strength of each cause’s contribution to each sample**. According to the results, causes like $c_1$  consistently demonstrate a strong positive influence on anomaly detection, suggesting that their representations are closely linked to anomaly-indicating patterns (e.g., "hardware failure" in a cloud service context). Conversely, causes like  $c_{7}$ ​ tend to shift predictions toward normal behavior, indicating that these causes are more reflective of regular instances (e.g., "user misperception"). Figures 7a and 7c illustrate the contributions of a specific sample, while Figure 7b provides an overview across all samples. We have included this friendly version's explanation in our revised paper (**Lines 460-500**). Thank you again for your valuable suggestion, which has greatly helped to improve the clarity of our paper!

---

> ### Author Response · Authors · 2024-11-16
> **Response to Question 1 and 2 by Authors**
>
> **[Questions]**
>
> **Q1.** If some non-causal factors can also be considered as causal factors depending on the domain?
>
> **A1.** Thank you for the insightful question. In the proposed SCM, the non-causal factors  $\boldsymbol{U}$  and the causal factors  $\boldsymbol{C}$  are defined as unobserved latent variables that represent a range of potential influences. Based on whether a factor directly causes $\boldsymbol{y}$ or only affects $\boldsymbol{X}$ without impacting $\boldsymbol{y}$, we can categorize it as either a causal factor $\boldsymbol{C}$ or a non-causal factor $\boldsymbol{U}$. This distinction is, therefore, *flexible and may vary depending on the specific domain or scenario*, exemplified by the additional healthcare example discussed in our answer for **W1**. This adaptability allows our model to accommodate different domains. We have added this discussion in **Appendix G.1**, and we really appreciate this valuable comment.
>
> **Q2.** What is the rationale for augmenting the raw input? Is it enough to simulate real-world disturbances? The performance of it? Would it have multiple ways of augmenting?
>
> **A2**. Thank you for the questions about the augmentation operation. We address your questions point by point:
>
> * **Rationale for augmentation**: As described in the first part of **Section 4.1**, the purpose of augmentation is to implement a causal intervention. Since $\boldsymbol{C}$ should be separated $\boldsymbol{U}$ according to the Common Cause Principle (**Section 3.2**), we perform an intervention on $\boldsymbol{U}$ does not make changes to $\boldsymbol{C}$. Then, by encouraging similarity between representations with and without $\boldsymbol{U}$, we aim to learn only the causal component, $\boldsymbol{C}$, as established in previous works [5,6].
> * **Simulating real-world disturbances**: We acknowledge that real-world scenarios can be more complex, and our approach aims to simulate the essential parts of this complexity. As described in **Lines 234-239**, based on insights from prior literature [7,8], non-causal elements frequently appear randomly, similar to noise typically found in the high-frequency components of time series data. Therefore, we perform causal intervention by adding noise to the high-frequency (less significant) components of the input data to simulate these non-causal disturbances while preserving the core causal structure.
> * **Performance impact**: As one of the core components of our framework, it indeed enhances the framework’s performance. This can be observed in our ablation study (**Table 2**), where the model without causal intervention (**w/o CI**) shows reduced performance, demonstrating the positive impact of causal intervention on our framework. We have also conducted additional experiments on more datasets, as shown in the following table, and hope to address your concern.
>
> | **Dataset**  | **Cloud-B**| **PSM**|
> |--------------|------------|--------|
> | **w/o CI**   | 0.888 $\pm$ 0.002 ($\downarrow$4.10\%)  | 0.726 $\pm$ 0.009 ($\downarrow$3.59\%) |
> | **w/o ICM**  | 0.889 $\pm$ 0.006 ($\downarrow$4.00\%)    | 0.725 $\pm$ 0.002 ($\downarrow$3.72\%) |
> | **w/o Attn** | 0.891 $\pm$ 0.002 ($\downarrow$3.78\%) | 0.723 $\pm$ 0.010 ($\downarrow$3.98\%)  |
> | **w/o GP**   | 0.894 $\pm$ 0.001 ($\downarrow$3.46\%) | 0.725 $\pm$ 0.009 ($\downarrow$3.72\%) |
> | **CaPulse**  | **0.926 $\pm$ 0.007**                  | **0.753 $\pm$ 0.042**                  |
>
> * **Multiple augment methods**: We agree with the suggestion to explore multiple augmentation methods. To address this, we have conducted additional experiments of different augmentation methods and combinations on two datasets (PSM and SMD), as shown in the following table.
> | Aug. Method  | PSM| SMD   |
> |--------|---------|--------|
> | HighFreq| 0.753 ± 0.042  | 0.906 ± 0.009  |
> | LowFreq| 0.743 ± 0.015  | 0.902 ± 0.007  |
> | Shift| 0.728 ± 0.011  | 0.885 ± 0.022  |
> | HighFreq + LowFreq| 0.725 ± 0.009  | 0.905 ± 0.005  |
> | HighFreq + Shift|  0.727 ± 0.011 | 0.884 ± 0.021  |
> | LowFreq + Shift|  0.725 ± 0.008 |  0.881 ± 0.018 |
> | HighFreq + LowFreq + Shift | 0.729 ± 0.014  | 0.874 ± 0.010  |
>
> **HighFreq** refers to adding noise to the high-frequency components, which was our initial approach. **LowFreq** denotes adding noise to the low-frequency components, and **Shift** indicates shifting the input time series by 20 time steps. The "+" symbol represents a combination of different methods. According to the results, adding noise to high-frequency components yields the best performance, with LowFreq also performing well but slightly below HighFreq. Shifting the time series has a lesser impact, and combining multiple augmentation methods does not improve performance beyond HighFreq alone, suggesting that excessive variability may obscure meaningful causal patterns. We have included this analysis in **Appendix G.2**, and we appreciate the valuable feedback that encouraged us to explore these augmentation strategies.

---

> ### Author Response · Authors · 2024-11-16
> **Response to Question 3 to 5 by Authors**
>
> **Q3.** How to determine an anomaly judgment? A threshold?
>
> **A4.** Yes, a threshold can indeed be applied to the anomaly score to classify instances as anomalous or normal. For evaluation, we follow previous density-based anomaly detection methods [9,10], using the ROC curve calculated from anomaly scores to evaluate our method.
>
> **Q4**. What is the meaning of Figure 6c?
>
> **A4.** Figure 6c provides a visualization of the flattened representations to help understand the structure of the cause pyramid representation. We have clarified this in the revised manuscript (**Line 452**). Thank you for your helpful question.
>
> **Q5**. In Figure 6a, I don't understand why CaPulse is the only one that can accurately predict the anomalies because I don't know why they are anomalies through the time-series plot.
>
> **A5**. Thank you for the question. Figure 6a demonstrates that CaPulse is uniquely capable of detecting anomalies because it identifies the true causal factors underlying these anomalies, rather than relying solely on patterns in the time-series data, which is exactly our motivation of the design of our framework. However, we admit that the anomaly labels do not specify the exact type of anomaly (e.g., hardware failure), and we also lack labels for potential user misoperations. Therefore, our analysis in this part is based on inferences drawn from observation and the results.
>
> We really appreciate your invaluable comments, which provided us with useful suggestions to make our work more solid and clearer. Your feedback means a great deal to us. We have revised our paper accordingly and hope that these changes address your questions and concerns. Thank you once again for your thoughtful input!
>
> **Reference**
>
> [1] Matrix profile I: all pairs similarity joins for time series: a unifying view that includes motifs, discords and shapelets. ICDM. 2016.
>
> [2] Breaking the Time-Frequency Granularity Discrepancy in Time-Series Anomaly Detection. WWW 2024.
>
> [3] TimesNet: Temporal 2D-Variation Modeling for General Time Series Analysis. ICLR 2023.
>
> [4] A dataset to support research in the design of secure water treatment systems. In International conference on critical information infrastructures security. 2016.
>
> [5] Causality Inspired Representation Learning for Domain Generalization. 2022.
>
> [6] Causal-Debias: Unifying Debiasing in Pretrained Language Models and Fine-tuning via Causal Invariant Learning. Proceedings of the 61st Annual Meeting of the Association for Computational Linguistics. 2023.
>
> [7] ​​Robusttad: Robust time series anomaly detection via decomposition and convolutional neural networks. 2021.
>
> [8] Through the dual-prism: A spectral perspective on graph data augmentation for graph classification. 2024.
>
> [9] Graph-augmented normalizing flows for anomaly detection of multiple time series. ICLR. 2022.
>
> [10] Detecting multivariate time series anomalies with zero known label. AAAI. 2023.

---

> ### Comment · Reviewer_kFnw · 2024-11-19
>
> Thank you for addressing the concerns I had with this paper.
> I recommend you revise the paper by adding all the experiments you said during the rebuttal period.
> My concerns have been addressed through rebuttal and I raise my score.

---

> > ### Author Response · Authors · 2024-11-19
> >
> > Thank you so much for your thoughtful feedback and for raising your score! We sincerely appreciate your recognition of the efforts we made during the rebuttal period. As suggested, we have already included some of the experiments and will ensure that all remaining experiments are incorporated into the revised paper. Your valuable input has truly improve the quality of our work, and we are grateful for your support. Thank you!

---

### Official Review · Reviewer_8nfA · 2024-11-01

**Soundness:** 2
**Presentation:** 3
**Contribution:** 1
**Rating:** 5
**Confidence:** 4

**Summary:**

The paper proposes a structural causal model (SCM) to understand the generation process of anomalies in time series data. CaPulse leverages causal tools to identify the true underlying causes of anomalies, enhancing interpretability and generalization capabilities. Additionally, it employs Periodical Normalizing Flows (PeNF) with a novel mask mechanism and attention mechanism to capture multi-period dynamics. This enables the model to effectively address the challenges of label scarcity, data imbalance, and complex multi-periodicity in TSAD.

**Strengths:**

S1. Time series anomaly detection is important to various domains.

S2. There are quite a few nice illustrations.

S3. This work focuses on an important problem that could have real-world applications.

S4. The figures and tables used in this work are clear and easy to read.

**Weaknesses:**

W1. The paper conducts ablation experiments solely on two datasets, as shown in Table 2. This narrow focus raises concerns about the generalizability of the findings. A more comprehensive analysis involving additional datasets could provide valuable insights into the method's performance and limitations.

W2. The approach presented appears to lack novelty, as it primarily builds upon established methods of causal inference and frequency domain analysis without offering significant advancements. Instead of innovating, the proposed method seems to merely combine existing techniques.

W3. The comparative analysis in Table 1 is limited, as the authors do not engage with the most advanced and relevant methods currently available in the literature. The selection of baseline models is inadequate, as it overlooks several cutting-edge techniques that could offer a more rigorous benchmark. To strengthen their evaluation, the authors should include comparisons with a wider array of state-of-the-art anomaly detection algorithms, thereby providing a clearer context for assessing the performance of their proposed method.

**Questions:**

Q1: Why were ablation experiments only conducted on two datasets in Table 2, and what were the effects on the other datasets?

Q2: Causal inference and frequency domain-based methods have already been proposed before. Your method doesn’t seem to have anything novel compared to existing methods. It seems like you are just combining them.

Q3: In Table 1, the methods you compare with are not the best current methods. There is a lack of comparison with the latest methods.

---

> ### Author Response · Authors · 2024-11-16
> **Response to Question 1 and 2 by Authors**
>
> We appreciate your thoughtful feedback and your recognition of the wide range of experiments and the novelty of our approach. We address your concern point by point as follows.
>
> **[Weaknesses]**
>
> Please see the answers to the **Questions** section.
>
> **[Questions]**
>
> **Q1**. The ablation experiments on the other datasets?
>
> **A1**. Thank you for your suggestion. Following your suggestion, we have conducted ablation studies on two additional datasets to further evaluate the generalizability of our approach. The results, presented in the tables below, show that removing any single component leads to noticeable performance drops, ranging from 3.46% to 4.1% on Cloud-B and 3.59% to 3.98% on PSM. In contrast, the full CaPulse model consistently achieves the highest performance. These findings underscore the critical contribution of each core component to the overall effectiveness of the model. Additionally, this analysis has been included in **Appendix F.1** in our revision for further reference.
>
> | **Dataset**  | **Cloud-B**                            | **PSM**                                |
> |--------------|----------------------------------------|----------------------------------------|
> | **w/o CI**   | 0.888 $\pm$ 0.002 ($\downarrow$4.10\%)  | 0.726 $\pm$ 0.009 ($\downarrow$3.59\%) |
> | **w/o ICM**  | 0.889 $\pm$ 0.006 ($\downarrow$4.00\%)    | 0.725 $\pm$ 0.002 ($\downarrow$3.72\%) |
> | **w/o Attn** | 0.891 $\pm$ 0.002 ($\downarrow$3.78\%) | 0.723 $\pm$ 0.010 ($\downarrow$3.98\%)  |
> | **w/o GP**   | 0.894 $\pm$ 0.001 ($\downarrow$3.46\%) | 0.725 $\pm$ 0.009 ($\downarrow$3.72\%) |
> | **CaPulse**  | **0.926 $\pm$ 0.007**                  | **0.753 $\pm$ 0.042**                  |
>
> **Q2**. Causal inference and frequency domain-based methods have already been proposed before. Concern about novelty.
>
> **A2**. Causal inference and frequency domain-based methods are broad fields encompassing a wide range of tasks and approaches. As discussed in **Section 2.2**, we have already reviewed relevant works and highlighted the gaps our approach aims to address. To further clarify, we have provided a comprehensive comparison with related work in the table below.
>
> |     | Method       |  Desciption   | Comparison with CaPulse         |
> |-------------------|---------------------|---------------------------------------------------|----------------------------------------------------------------------|
> |           TSAD methods           | GANF [1]                           | Models statistical dependencies   using graph structures and density estimation via normalizing flows.            |        GANF lack a causal perspective and multi-periodicity modeling. While CaPulse   capture generative processes and multiple periodics.                                                             |
> |                                  | MTGFlow [2]                        |        Entity-aware designs for robust multivariate time-series anomaly detection.                                |        MTGFlow does not account for causal mechanisms or multi-periodicity.                                                                                                                            |
> |                                  | AnomalyTransformer [3], DualTF [4] | Improved detection accuracy using reconstruction-based   models                                                   | Relies on statistical patterns without modeling causal processes. CaPulse integrates a causal view to   improve robustness and interpretability, handling noisy conditions via   density estimation. |
> | Causal inference   based methods | Cost [5]                           | Uses a contrastive learning framework to disentangle seasonal and trend   components for time series forecasting. | Cost addresses forecasting    probelm, whereas CaPulse is tailored for TSAD, emphasizing causal   mechanisms behind anomaly generation.                                                                |
> |                                  |        CaseQ [6]                   | Introduces a causal approach for predicting sequential events in   out-of-distribution settings.                  | CaseQ addresses sequential event prediction, whereas CaPulse is designed   for TSAD.                                                                                                                   |
>
> Based on this, we would like to restate our contribution as follows: We propose **CaPulse**, a causal framework for time series anomaly detection that identifies anomalies by capturing disruptions in **underlying generative processes** and **multi-period dynamics**, providing both high accuracy and **interpretability**.
>
> We have also added an extended discussion of the comparison in **Appendix G.3** of our revised submission, which we hope will address your concerns. If there are specific related works you believe we may have overlooked, please let us know, and we will be happy to provide further clarification.

---

> ### Author Response · Authors · 2024-11-16
> **Response to Question 3 by Authors**
>
> **Q3**. Lack of comparison with the latest methods.
>
> **A3**. Thank you for your valuable feedback. To address your concern, we plan to add three additional baselines, including two recent baselines (DualTF [7] and TimesNet [8]) and one classical method (MatrixProfile [9]). Due to time constraints, we commit that all experiments will be conducted in full, and be included in our final revision.
>
>
> |                     | **SWaT**        | **WADI**        | **PSM**         | **SMD**         |
> |---------------------|-----------------|-----------------|-----------------|-----------------|
> | **MatrixProfile** | 0.600         | 0.677         | 0.634         | 0.866         |
> | **DualTF**        | 0.769 ± 0.019 | 0.796 ± 0.030 | 0.727 ± 0.071 | 0.796 ± 0.101 |
> | **TimesNet**      | **0.789 ± 0.016** | 0.756 ± 0.013 | 0.743 ± 0.029 | 0.882 ± 0.010 |
> | **CaPulse**         | 0.782 ± 0.004 | **0.830 ± 0.029**   | **0.753 ± 0.042**   | **0.906 ± 0.009**   |
>
> We greatly value your feedback and have included the necessary revisions in our updated version. We hope our responses clarify your concerns. **As these do not pertain to fundamental technical issues, we kindly request you to reconsider our score**. Thank you for your time and thoughtful review!
>
> **Reference**
>
> [1] Graph-Augmented Normalizing Flows for Anomaly Detection of Multiple Time Series. ICLR. 2022.
>
> [2] Detecting Multivariate Time Series Anomalies with Zero Known Label. AAAI. 2023.
>
> [3] Anomaly Transformer: Time Series Anomaly Detection with Association Discrepancy. ICLR. 2022.
>
> [4] Breaking the Time-Frequency Granularity Discrepancy in Time-Series Anomaly Detection. WWW. 2024.
>
> [5] CoST: Contrastive Learning of Disentangled Seasonal-Trend Representations for Time Series Forecasting. ICLR. 2022.
>
> [6] Towards Out-of-Distribution Sequential Event Prediction: A Causal Treatment. NeurIPS. 2022.
>
> [7] Breaking the Time-Frequency Granularity Discrepancy in Time-Series Anomaly Detection. WWW 2024.
>
> [8] TimesNet: Temporal 2D-Variation Modeling for General Time Series Analysis. ICLR 2023.
>
> [9] Matrix profile I: all pairs similarity joins for time series: a unifying view that includes motifs, discords and shapelets. ICDM. 2016.

---

> > ### Author Response · Authors · 2024-11-24
> > **Kindly Request for Feedback of Reviewer**
> >
> > Dear Reviewer 8nfA,
> >
> > As the rebuttal deadline is coming soon, please let us know if our responses have addressed your main concerns. If so, we kindly ask for your reconsideration of the score. If any aspects require additional elaboration or refinement, we will be more than happy to engage in further improvements and discussion.
> >
> > Thanks again for your time.

---

> > > ### Comment · Reviewer_8nfA · 2024-11-26
> > > **I have read the rebuttal and revised paper.**
> > >
> > > I have read the rebuttal and revised paper. However, I believe this paper is more of an application of existing theories to a new domain, representing an incremental improvement. Furthermore, i also agree on the comments from Reviewer 9PkW, that there are still some flaws in the theoretical parts. Therefore, I think my current score is fair. Best regards.

---

> > > > ### Author Response · Authors · 2024-11-27
> > > >
> > > > Thank you for reviewing our rebuttal and revised paper. We appreciate your feedback and the time you have taken to assess our work.
> > > >
> > > > While we acknowledge that incremental improvements are a vital part of scientific progress, our motivation for this work is rooted in the belief that introducing a new perspective to an existing problem adds meaningful value to the field.
> > > > Regarding the theoretical aspects, we note that no specific issues were raised in your earlier comments. However, if you have particular concerns in mind now, we would be sincerely grateful if you could share them to help us further improve the quality and clarity of our paper. For the issue that Reviewer 9PkW raised, please refer to our response to their questions.
> > > > Thank you again for your thoughtful review.
> > > >
> > > > Best regards,
> > > >
> > > > The Authors

---

### Official Review · Reviewer_q2Qb · 2024-11-03

**Soundness:** 2
**Presentation:** 2
**Contribution:** 2
**Rating:** 5
**Confidence:** 4

**Summary:**

The paper proposes CaPulse a method for time series anomaly detection that leverages causal inference. It introduces a structural causal model to understand anomaly generation, combines it with periodical normalizing flows for density estimation. The paper purports to address key challenges including label scarcity, data imbalance, and multiple periodicities.

---

Update: I appreciate the author's rebuttal that addressed a lot of my concerns, in particular around baselines, evaluation, and benchmarks. However, I agree with professor Keogh's point regarding the mischaracterization of MP and fail to understand the author's statement that "MP-based methods fall outside the primary scope of our paper" and that the focus of the paper in on Deep-learning. This does not appear to be the initially defined scope of the paper.

**Strengths:**

The paper's main strength is its integration of causal inference with time series anomaly detection and proposed solution for handling multiple periodicities. This method offers some degree of interpretability through SHAP. The authors do establish strong theoretical foundation for their method.

**Weaknesses:**

In my view the main weakness is with the empirical evaluation. The proposed method is extremely complex, likely computationally demanding with a large number of hyperparameters. The authors do perform some sensitivity analysis but it is limited. The baselines used for the empirical comparison relies exclusively on similarly complex baselines. The authors do not compare to simple, algorithmic baselines or methods such as Matrix Profile which have proven to outperform state of the art at a computational cost several orders of magnitude smaller. Without such comparisons, it is impossible to assess whether the complexity of the method is justified.

**Questions:**

How does the computational complexity scale with time series length and dimensionality?
What are the computational requirements during training and inference?
How is the optimal hyperparameter configuration (Appendix E) for each dataset established?
Can you provide a comparison with more parameter-efficient baselines?

---

> ### Author Response · Authors · 2024-11-16
> **Response to Weakness by Authors**
>
> Thank you for your insightful comments and for recognizing the clarity and experimental design of our approach. We address each of your concerns below.
>
> **[Weaknesses]** The proposed method appears computationally intensive with many hyperparameters, and the empirical comparison focuses only on similarly complex baselines, neglecting simpler algorithmic methods.
>
> Thank you for raising your concern regarding the computational complexity of our approach. We address your points in detail as follows:
>
> **A1. Theoretical computational complex**: As discussed in **Appendix B.3**, the theoretical complexity of our approach is $\mathcal{O}(T \log T) + \mathcal{O}(N^2 D_h)$. The first term, $\mathcal{O}(T \log T)$, results from the FFT used to extract global and local periods, with the global period calculated as a preprocessing step. The second term, $\mathcal{O}(N^2 D_h)$, corresponds to the attention mechanism in the MpCF module. Notably, the PeNFs are linear, contributing minimal additional complexity.
>
> **A2. Efficiency comparison**: Theoretically, the complexity of Matrix Profile (MP) is  $\mathcal{O}(T_l^2 \log T_l) $ [1], where $T_l$ represents the total length of the time series (typically, $T_l \gg T$). Thus **MP’s theoretical complexity is higher** than that of our approach. To further address your concern, we conducted experiments on four datasets and measured the time costs. However, we think that a direct efficiency comparison may be unfair for several reasons:
>
> 1. Methods like MP can only be run on the CPU, while DL methods such as CaPulse can leverage GPU acceleration.
> 2. MP operates directly on the test data, which is smaller (about one-third of the training set size), whereas CaPulse is trained on the full training set.
> 3. Training epochs vary across datasets and can be adjusted, making the total training time flexible.
>
> Thus, to provide additional context, we also included a comparison with a recent DL-based method, DualTF [2].
>
>   | **Dataset** | **Metric**      | **MatrixProfile** | **DualTF**             | **CaPulse**            |
> |-------------|-----------------|-------------------|------------------------|------------------------|
> | **PSM**     | **# Param (k)** | -                 | 4801.6                | 204.7                 |
> |             | **Time cost**   | 25.944 (s)        | 2.265 ± 0.356 (s/epoch)| 0.533 ± 0.192 (s/epoch)|
> |             | **ROC**         | 0.634             | 0.727 ± 0.071         | 0.753 ± 0.042          |
> | **SMD**     | **# Param (k)** | -                 | 4820                  | 264.7                 |
> |             | **Time cost**   | 24.673 (s)        | 0.709 ± 0.385 (s/epoch)| 0.182 ± 0.195 (s/epoch)|
> |             | **ROC**         | 0.866             | 0.796 ± 0.101         | 0.906 ± 0.009          |
> | **WADI**    | **# Param (k)** | -                 | 4949.1                | 342.2                 |
> |             | **Time cost**   | 40.428 (s)        | 4.52 ± 0.372 (s/epoch) | 2.505 ± 0.197 (s/epoch)|
> |             | **ROC**         | 0.677             | 0.796 ± 0.030         | 0.830 ± 0.029          |
> | **SWaT**    | **# Param (k)** | -                 | 4840.5                | 242.4                 |
> |             | **Time cost**   | 43.065 (s)        | 11.244 ± 0.34 (s/epoch)| 3.613 ± 0.243 (s/epoch)|
> |             | **ROC**         | 0.600             | 0.769 ± 0.019         | 0.782 ± 0.004          |
>
> Based on the table, we observe that CaPulse achieves significantly lower time costs per epoch and consistently outperforms Matrix Profile and DualTF in ROC, demonstrating both **efficiency** and **effectiveness**.
>
> **A3. Performance comparison with classical methods**: Thank you for your suggestion to include more classical baselines. Following your advice, we have added three additional baselines — MP, KNN, and K-means — to our evaluation, as shown below. The results indicate that CaPulse consistently achieves superior ROC scores compared to classical methods, reinforcing its robustness and accuracy in detecting anomalies across diverse datasets.
>
> |        | **SWaT**        | **WADI**        | **PSM**        | **SMD**        |
> |-------------------|-----------------|-----------------|----------------|----------------|
> | **MatrixProfile** | 0.600           | 0.677           | 0.634          | 0.866          |
> | **KNN**           | 0.716           | 0.815           | 0.654          | 0.496          |
> | **Kmeans**        | 0.560           | 0.639           | 0.535          | 0.692          |
> | **CaPulse**       | 0.782 ± 0.004   | 0.830 ± 0.029   | 0.753 ± 0.042  | 0.906 ± 0.009  |
>
> Thank you for your valuable comments. We have included this discussion and experimental results in **Appendix F.2** of our revised paper.
>
> [1] Matrix profile I: all pairs similarity joins for time series: a unifying view that includes motifs, discords and shapelets. ICDM. 2016.
>
> [2] Breaking the Time-Frequency Granularity Discrepancy in Time-Series Anomaly Detection. WWW. 2024.

---

> ### Author Response · Authors · 2024-11-16
> **Response to Questions by Authours**
>
> **[Questions]**
>
> **Q1.** How does the computational complexity scale with time series length and dimensionality?
>
> **A1.** Thanks for your question. The information you requested has already been included in **Appendix B.3** of our original paper. Specifically, the theoretical complexity of our approach is $\mathcal{O}(T \log T) + \mathcal{O}(N^2 D_h)$. The first term, $\mathcal{O}(T \log T)$, results from the FFT used to extract local periods. The second term, $\mathcal{O}(N^2 D_h)$, corresponds to the attention mechanism in the MpCF module. Notably, the PeNFs are linear, contributing minimal additional complexity.
>
> **Q2&Q4**. What are the computational requirements and the comparison with parameter-efficient baselines?
>
> **A2**. Please see the answer for weakness.
>
> **Q3**. What is the optimal hyperparameter configuration?
>
> **A3**. Thank you for your questions, and we apologize for missing the details of the implementation. For each method, we employed publicly available implementations and configured hyperparameters for fair comparisons. The hidden dimension was uniformly set to 64 for most methods, with two layers in their network architecture, unless otherwise specified, e.g., adversarial methods like DROCC and ALOCC used additional parameters such as gamma (set to 2) and lambda (set to 0.0001) in DROCC, while USAD included specific configurations for $\alpha$ and $\beta$, both set to 0.5. Transformer-based approaches like AnomalyTransformer had a window size of 60, 8 attention heads, and a feedforward network dimension of 512. We have accordingly included these details in our revision of **Appendix D**. We hope this addresses your concerns, and thank you again for pointing this out.
>
> Thank you for your valuable comments. We have addressed them in our revised submission and hope our responses resolve your concerns. **Since these are not critical technical flaws, we sincerely ask for a reassessment of our score.** We truly appreciate your review and consideration!

---

> > ### Author Response · Authors · 2024-11-25
> > **Kindly Request for Reviewer's Feedback**
> >
> > Dear Reviewer,
> >
> > Thank you so much for your time in improving our paper!
> >
> > Since the end of the rebuttal is coming very soon, may we know if our response addresses your main concerns? If so, we kindly ask for your reconsideration of the score. Should you have any further advice, please let us know and we will be more than happy to engage in more discussion and improvements.

---

> > > ### Author Response · Authors · 2024-12-02
> > > **Respectful Inquiry Before Discussion Deadline**
> > >
> > > Dear reviewer q2Qb,
> > >
> > > Thank you for taking the time and effort to provide a valuable review of our work. As we are approaching the end of the discussion, we hope that you have had the chance to review our previous response. If our response has addressed your concerns, we thank you for reconsidering the score, and we are more than willing to engage in further discussion if needed.
> > >
> > > Yours sincerely,
> > >
> > > Authors

---

### Official Review · Reviewer_9PkW · 2024-11-03

**Soundness:** 2
**Presentation:** 1
**Contribution:** 2
**Rating:** 3
**Confidence:** 4

**Summary:**

This paper proposes an anomaly detection framework that combines several existing algorithmic building blocks, including normalized flows, FFTs, and patch-based masking. The overall objective is to maximize the log-likelihood, which appears to produce a latent representation attributed to causal factors.

**Strengths:**

The paper makes an effort to integrate multiple algorithmic building blocks into a learning system.

**Weaknesses:**

It seems that the learning problem could benefit from a clearer explanation. The objective function appears to be conditioned by $ C_\text{ind} $ and $C_0$, but the criteria for selecting these values are not immediately obvious. This aspect is essential, especially when considering unsupervised learning tasks with latent variables. It is possible that a two-stage optimization strategy, where $C$ parameters and other model parameters are optimized in an alternating fashion, has been implemented, though a detailed description of this approach does not seem to be readily available. At the very least, it would be helpful if the parameters to be learned were clearly specified.

In my attempt to understand the algorithm’s operation and the rationale behind its design choices, I encountered a few challenging aspects. These include, for example, the use of FFT in Eq. (1), the orthogonality condition on $ C_\text{ind} $ and its connection to causal learning, and the role of the "pyramid" structure, which may be intended for multi-scale convolution across spatial and temporal dimensions. There are various possible approaches for identifying independent causal factors or periodic patterns at different granularities. In general, a well-written paper typically provides some technical rationale when selecting a specific approach. At present, there appears to be a slight disconnect between the authors’ intended objectives and the selected algorithmic components. For instance, it remains somewhat unclear why the resulting subspaces would lend themselves to causal interpretation or how the noise injection approach contributes to distinguishing confounders.

Given these considerations, I find it somewhat challenging to fully assess the framework’s novelty at this stage. I am inclined to suggest that additional development and clarification may be beneficial for the paper to reach its full potential for publication.

**Questions:**

Please address what described in weakness.

---

> ### Author Response · Authors · 2024-11-16
> **Response by Authors**
>
> We sincerely appreciate your thoughtful comments. Below, we address the concerns you have raised in detail.
>
> **[Weakness]**
>
> **W1. The objective function’s detail is unclear.**
>
> **A1.** Regarding the objective function, we believe there may be a misunderstanding, and we would like to clarify the following points:
>
> 1. The objective function is **conditioned solely on $C_{ind}$** and not on $C_{o}$ (an intermediate result), as outlined in **Eq. 3** and **5**.
> 2. We do not explicitly select $C_{ind}$; rather, it is learned automatically through the PaCM and MpCF modules, as illustrated in **Figure 3** of our framework. This learning process is also detailed in **Section 4.2** and **Appendix B.1**.
> 3. Our approach is not a two-stage optimization strategy. As stated in **Section 4.4**, the total loss is defined as: $\mathcal{L} = \mathcal{L}{\text{nf}}+\alpha \mathcal{L}{\text{sim}} + \beta\mathcal{L}{\text{ind}}$, indicating the model is trained in an **end-to-end** manner.
>
> Thank you for your comments, we have revised the manuscript to ensure that these details are more easily accessible and explicitly highlight the parameters to be learned, as suggested (**Line 328-329**).
>
> **W2. Some algorithms’ operations and rationales** **behind the design choice are unclear.**
>
> **A2.** We appreciate your question and make the following clarification:
>
> 1. **Use of FFT**: Thank you for raising this point. As described in **Lines 233–235**, we use FFT to perform causal intervention by adding noise to the less significant parts of the data. FFT is a widely used method for separating high- and low-frequency components, enabling targeted intervention on high-frequency noise. We have clarified this in our revision (**Lines 242–243**).
> 2. **Orthogonality and causality**: We employ orthogonality to satisfy the Independent Causal Mechanisms assumption, which requires none of the factorizations of C to entail information from others, thus it enforces the mutual independence of the causal factors. This is described in **Lines 186 - 189**.
> 3. **Pyramid structure**: The pyramid structure is specifically designed to capture multi-period information, as highlighted in the introduction (**Lines 90–91**).
> 4. **Noise injection approach**: As described in **Lines 234-239**, based on insights from prior literature [1,2], non-causal elements frequently appear randomly, similar to noise typically found in the high-frequency components of time series data. Therefore, we perform causal intervention by adding noise to the high-frequency (less significant) components of the input data to simulate these non-causal disturbances while preserving the core causal structure. We acknowledge that real-world scenarios can be more complex, thus we have conducted additional experiments of different augmentation methods and combinations on two datasets (PSM and SMD), as shown in the following table.
>
> | Augmentation Method      | PSM            | SMD            |
> |----------------------------|----------------|----------------|
> | HighFreq                   | 0.753 ± 0.042  | 0.906 ± 0.009  |
> | LowFreq                    | 0.743 ± 0.015  | 0.902 ± 0.007  |
> | Shift                      | 0.728 ± 0.011  | 0.885 ± 0.022  |
> | HighFreq + LowFreq         | 0.725 ± 0.009  | 0.905 ± 0.005  |
> | HighFreq + Shift           |  0.727 ± 0.011 | 0.884 ± 0.021  |
> | LowFreq + Shift            |  0.725 ± 0.008 |  0.881 ± 0.018 |
> | HighFreq + LowFreq + Shift | 0.729 ± 0.014  | 0.874 ± 0.010  |
>
>
> **HighFreq** refers to adding noise to the high-frequency components, which was our initial approach. **LowFreq** denotes adding noise to the low-frequency components, and **Shift** indicates shifting the input time series by 20 time steps. The "+" symbol represents a combination of different methods. According to the results, adding noise to high-frequency components yields the best performance, with LowFreq also performing well but slightly below HighFreq. Shifting the time series has a lesser impact, and combining multiple augmentation methods does not improve performance beyond HighFreq alone, suggesting that excessive variability may obscure meaningful causal patterns. We have included this discussion in **Appendix G.2**.
>
> We sincerely appreciate your thoughtful comments and hope our responses have adequately addressed your concerns. If there is anything we haven’t fully explained, please don’t hesitate to let us know—we are more than happy to provide further clarification. **As these points are not fundamentally technical issues, we kindly ask you to reconsider our score.** Thank you very much for your time and consideration!
>
> **Reference**
>
> [1] ​​Robusttad: Robust time series anomaly detection via decomposition and convolutional neural networks. 2021.
>
> [2] Through the dual-prism: A spectral perspective on graph data augmentation for graph classification. 2024.

---

> > ### Comment · Reviewer_9PkW · 2024-11-25
> >
> > I have reviewed the revised paper and found that the issues I initially raised remain unaddressed. Specifically, the authors have not clarified the connection between geometrical orthogonality in Euclidean space and the independence of causal factors. Furthermore, the fundamental question regarding the problem setting of the learning problem is still unresolved.
> >
> > The paper includes several operations that appear to be selected arbitrarily, lacking clear explanations or justifications. Consequently, I believe that this paper does not make sufficient contribution to the machine learning community due to its lack of clarity. Therefore, I am convinced that it is not yet ready for publication in its current form at ICLR.
> >
> > However, it might be able to make a valuable contribution in a community where algorithmic details are less critical. I suggest resubmitting the paper to another venue.

---

> > > ### Author Response · Authors · 2024-11-27
> > >
> > > Thank you for your response to our rebuttal and for highlighting areas where our work could be further improved. We deeply appreciate your feedback and would like to provide additional clarification regarding the concerns you raised:
> > >
> > > 1. **Orthogonality loss for Independence**: We are here to justify the use of orthogonality loss for joint independence of $ C = \lbrace c_1, c_2, \ldots, c_N \rbrace  \in \mathbb{R}^{N \times D_c} $ (Line 162). If $ c_i $ and $ c_j $ are independent random variables, then their expectations satisfy the relation: $\mathbb{E}[c_{i}c_{j}] = \mathbb{E}[c_i]\mathbb{E}[c_j]$. Orthogonality is defined as: two objects being orthogonal if their inner product is zero. In the context of random variables, the inner product can be defined when the variables are square-integrable, i.e., they satisfy: $\mathbb{E}[c_i^2] &lt; \infty \quad \text{and} \quad \mathbb{E}[c_j^2] &lt; \infty$, which is satisfied in our context. For such random variables, the inner product is given by: $(c_i, c_j) := \mathbb{E}[c_{i}c_{j}]$. This definition aligns with the requirements of the Cauchy-Schwarz inequality, which ensures that this inner product satisfies the properties of a valid geometric structure. Now, consider $ X $ and $ Y $ after subtracting their means, so that: $\mathbb{E}[c_{i}] = 0 \quad \text{and} \quad \mathbb{E}[c_{j}] = 0$. By construction: $\mathbb{E}[c_{i}]\mathbb{E}[c_{j}] = 0$. If $ X $ and $ Y $ are orthogonal (i.e., $\mathbb{E}[c_{i}c_{j}] = 0 $), then their inner product vanishes: $\mathbb{E}[c_{i}c_{j}] = 0$. In this setup, orthogonality implies that the variables guarantee that $ \mathbb{E}[c_{i}c_{j}] = \mathbb{E}[c_{i}]\mathbb{E}[c_{j}] $.
> > > 2. **Problem Setting**: Our primary learning goal is **density-based anomaly detection**. As noted in Lines 333–338 of our manuscript, we follow prior works [1, 2] to use the negative logarithm of the likelihood of the input time series as the anomaly score. If you have more specific questions or concerns, we would be happy to address them further.
> > > 3. **Contribution to the Community**: We would like to restate that the main contribution of our work lies in the **causal view** presented, which relates and extends concepts explored in many prior accepted ICLR papers [3–5]. This causal perspective forms the core of our contribution to the ICLR community.
> > >
> > > We hope that these clarifications address your concerns. Thank you again for your time.
> > >
> > > Best regards,
> > >
> > > Authors

---

### Author Response · Authors · 2024-11-16
**Global Response**

Dear Reviewers,

We would like to express our sincere gratitude to all the reviewers for their constructive feedback on our manuscript. Your insights have been invaluable in improving the quality and clarity of our work. We also sincerely thank you for acknowledging the **novelty** **of our approach** (R4), the **clarity of our presentation and illustrations** (R3), and the **comprehensive interpretability analysis** (R2, R4).

Following your suggestions, we have made several revisions to address the concerns raised:

* **Enhanced Clarifications**: We have provided clearer explanations for key aspects, including the objective functions (R1W1), experimental settings (R2Q3), interpretability explanation (R4W3), and design rationale (R1W2, R4Q2).
* **Additional Experiments**: To further validate our approach, we conducted additional experiments. These include additional ablation studies (R3Q1), more baselines (R3Q3, R4W2), and efficiency comparisons (R2W2).
* **Related Works Discussion**: We have enriched our discussion and comparison of related work to better position our contributions within the existing body of research (R3Q2).

These revisions have been included in our revised paper (**Section 5.3**, **Appendix F** and **Appendix G**). We believe these revisions have significantly enhanced the manuscript. We hope that our responses and changes address your concerns satisfactorily.

To ensure clarity, we provide the reviewer identifiers as follows:
* Reviewer 9PkW is referred to as R1.
* Reviewer q2Qb is referred to as R2.
* Reviewer 8nfA is referred to as R3.
* Reviewer kFnw is referred to as R4.

Once again, thank you for your time and effort in reviewing our work. We look forward to your continued feedback!

Best regards,

Authors

---

> ### Author Response · Authors · 2024-11-22
> **Updated Revised Manuscript with All Additional Results**
>
> Dear Reviewers,
>
> We would like to inform you that we have just completed all experiments and have further updated our revised manuscript. Specifically, we have included results from **five additional baselines in Table 1**, as per your suggestion: three traditional methods (MP, KNN, and KMeans) and two recent approaches (DualTF and TimesNet), evaluated across all datasets.
>
> Additionally, we would like to kindly remind you that **the discussion phase is ending soon on Nov. 26th**. We sincerely hope reviewers who have not yet had the opportunity to review our initial response can take a look at our response and the revised manuscript and ensure that all concerns have been addressed.
>
> We greatly appreciate your constructive feedback and remain available to provide further clarifications or make additional refinements if needed. Thank you for your time and effort in reviewing our work!
>
> Best regards,
>
> Authors

---

### Public Comment · ~Eamonn_Keogh1 · 2024-11-23

The claim "Thus MP’s theoretical complexity is higher than that of our approach." is not correct.

For the anomaly detection, you don't need the full MP, just the highest points, the discords
You can do that at `100,000 Hz with ease, even on a cheap computer.

In fact, paper [a] looks at datasets of lengths unto a trillion data points!
--
You test on "PSM SMD MSL", but in [b] there is strong evidence that you cannot make meaningful claims on these datasets.

Best wishes with your work

[a] Matrix Profile XXIV:Scaling Time Series Anomaly Detection to Trillions of Datapoints and Ultra-fast Arriving Data Streams. Yue Lu, Renjie Wu, Abdullah Mueen, Maria A. Zuluaga and Eamonn Keogh. ACM SIGKDD 2022

[b] https://www.dropbox.com/scl/fi/cwduv5idkwx9ci328nfpy/Problems-with-Time-Series-Anomaly-Detection.pdf?rlkey=d9mnqw4tuayyjsplu0u1t7ugg&dl=0

---

> ### Author Response · Authors · 2024-11-23
>
> Dear Prof. Eamonn Keogh,
>
> Thank you for your valuable comments. We really appreciate the contributions of Matrix Profile (MP) in time series data mining, and would like to provide the following clarification:
>
> - Computational Complexity: As suggested by Reviewer q2Qb, we have included the comparison with the MP method [1] in our revision. The theoretical computational complexity of MP we stated in our response is referred from [1], as stated in the first line of Section III.C in the original paper: “_The overall complexity of the proposed algorithm is $\mathcal{O}(n^{2} log n)$, where n is the length of the time series_”. In contrast, the complexity of our method is $\mathcal{O}(T \log T) + \mathcal{O}(N^2 D_h)$. Based on this distinction, we made the claim regarding complexity in our work. Thanks to your comments, we will carefully review the new literature [2] you provided to explore potential improvements in the implementation of MP.
>
> - Our Scope and Focus: Recently, deep learning methods have achieved superior performance in TSAD, compared to conventional approaches (e.g., KNN) [3-4]. Following this trend, our work is centered on **Deep Learning** (DL) for TSAD with an emphasis on accuracy, which guided our selection of baselines [5–12]. As such, **MP-based methods fall outside the primary scope of our paper**, as well as the papers in the DL-based TSAD literature. To ensure fairness, we have considered comparisons of both efficiency and effectiveness with DL-based approaches. In our response to Rq2Qb’s W2, we explained why we believe a direct comparison between DL-based methods and MP-like approaches is not ideal. However, to address the concern raised, we still have included this comparison in our work.
>
> Thank you once again for your comments.
>
> Best regards,
>
> Authors
>
> Reference:
>
> [1] Matrix profile I: all pairs similarity joins for time series: a unifying view that includes motifs, discords and shapelets. ICDM. 2016.
>
> [2] Matrix Profile XXIV:Scaling Time Series Anomaly Detection to Trillions of Datapoints and Ultra-fast Arriving Data Streams. Yue Lu, Renjie Wu, Abdullah Mueen, Maria A. Zuluaga and Eamonn Keogh. ACM SIGKDD 2022
>
> [3] Deep Learning for Time Series Anomaly Detection: A Survey. 2024.
>
> [4] A Comparative Analysis of Traditional and Deep Learning-based Anomaly Detection Methods for Streaming Data. 2019.
>
> [5] Breaking the Time-Frequency Granularity Discrepancy in Time-Series Anomaly Detection. WWW 2024.
>
> [6] TimesNet: Temporal 2D-Variation Modeling for General Time Series Analysis. ICLR 2023.
>
> [7] Graph-augmented normalizing flows for anomaly detection of multiple time series. ICLR. 2022.
>
> [8] Detecting multivariate time series anomalies with zero known label. AAAI. 2023.
>
> [7] Anomaly Transformer: Time Series Anomaly Detection with Association Discrepancy. ICLR. 2022.
>
> [9] USAD : UnSupervised Anomaly Detection on Multivariate Time. KDD. 2020
>
> [10] Deep autoencoding gaussian mixture model for unsupervised anomaly detection. ICLR. 2018.
>
> [11] Deep semi-supervised anomaly detection. ICLR. 2020.
>
> [12] Towards a Rigorous Evaluation of Time-series Anomaly Detection. AAAI. 2022

---

> > ### Public Comment · ~Eamonn_Keogh1 · 2025-02-26
> >
> > Hello.
> > The formatting made some of my text appear very large, as if I was "shouting" , not my attention, sorry (I don't understand the formatting tags)
> >
> > " As such, MP-based methods fall outside the primary scope of our paper, as well..."
> > Understood.
> >
> > I have pointed out that "SWaT PSM SMD MSL SMD" do not allow for meaningful comparisons for several years now [a].
> > It would be so much better for the community when we kill these datasets ;-)
> >
> > Best wishes
> >
> >
> > [a] https://www.dropbox.com/scl/fi/cwduv5idkwx9ci328nfpy/Problems-with-Time-Series-Anomaly-Detection.pdf?rlkey=d9mnqw4tuayyjsplu0u1t7ugg&dl=0

---

### Meta-Review · Area_Chair_VisW · 2024-12-19

**Metareview:**

Based on the reviews, I conclude that the paper cannot be accepted for publication in its current form. Of the four reviews, three recommend rejection, raising significant concerns about key aspects of the work, particularly the evaluation methodology. As an expert in the topic, I strongly agree with this assessment. Furthermore, the authors’ responses are evasive and fail to adequately address the reviewers’ concerns.

**Additional Comments On Reviewer Discussion:**

The key issues raised by the reviewers, along with the public comment, focused on clarity, novelty, theoretical grounding, and empirical evaluation:

- **Clarity and Novelty**: Reviewer 9PkW criticized the unclear explanation of the algorithm and its design rationale, as well as the connection between orthogonality and causal factor independence. Despite revisions, these concerns were not fully addressed, leading the reviewer to question the paper’s novelty and contribution.

- **Empirical Evaluation**: Reviewers q2Qb and 8nfA raised concerns about the complexity of the proposed method, limited baseline comparisons, and lack of sensitivity analysis. While the authors added new baselines and presented preliminary results, the rebuttal failed to resolve the reviewers' concerns.

- **Theoretical Foundation**: Reviewer 8nfA felt the paper offered incremental advances rather than novel contributions, while Reviewer 9PkW also found the theoretical aspects unclear. Public comment from Eamonn Keogh further emphasized that claims about the complexity of MP-based methods were incorrect.

- **Rebuttal Responses**: The authors attempted to address concerns by adding experiments and clarifications, but key issues regarding the clarity of the methodology, theoretical novelty, and empirical rigor remained unresolved.

In conclusion, despite the authors' revisions, the paper did not adequately address the major concerns raised by the reviewers, particularly regarding its contribution to the field and empirical evaluation, leading to a recommendation for rejection.

---

### Decision · Program_Chairs · 2025-01-22

Reject